# Gradient Descent Converges Arbitrarily Fast
# for Logistic Regression via Large and Adaptive Stepsizes

**Ruiqi Zhang**[1]  **Jingfeng Wu**[1]  **Peter L. Bartlett**[1][2]

## Abstract

We analyze the convergence of gradient descent (GD) with large, adaptive stepsizes for logistic regression on linearly separable data. The stepsize adapts to the current risk, scaled by a fixed base stepsize $\eta$. We prove that once the number of iterates $t$ surpasses a margin-dependent threshold, the averaged GD iterate achieves a risk upper bound of $\exp(-\Theta(\eta t))$, where $\eta$ can be chosen arbitrarily large. This implies that GD attains *arbitrarily fast* convergence rates via large stepsizes, although the risk evolution might not be monotonic. In contrast, prior adaptive stepsize GD analyses require a monotonic risk decrease, limiting their rates to $\exp(-\Theta(t))$. We further establish a margin-dependent lower bound on the iteration complexity for any first-order method to attain a small risk, justifying the necessity of the burn-in phase in our analysis. Our results generalize to a broad class of loss functions and two-layer networks under additional assumptions.

## 1. Introduction

*Gradient descent* (GD) and its variants are a popular class of optimization methods in modern machine learning and deep learning. In this paper, we study the convergence of GD with *adaptive stepsizes* given by

$$\mathbf{w}_0 = 0, \quad \mathbf{w}_{t+1} := \mathbf{w}_t - \eta_t \nabla \mathcal{L}(\mathbf{w}_t), \quad t \geq 0, \quad \text{(GD)}$$

where $\mathcal{L}(\cdot)$ is the loss objective to be minimized, $\mathbf{w}_t \in \mathbb{R}^d$ is the trainable parameters, and $\eta_t > 0$ is the stepsize at the $t$-th step. In particular, we allow $\eta_t$ to be a function of the current risk $\mathcal{L}(\mathbf{w}_t)$. Setting the initialization to zero does not cause the loss of generality.

[1]University of California, Berkeley, USA [2]Google DeepMind, USA. Correspondence to: Ruiqi Zhang <rqzhang@berkeley.edu>.

This work is extended in https://arxiv.org/abs/2504.04105, and Licong Lin was included as a co-author.

*Proceedings of the 42$^{nd}$ International Conference on Machine Learning*, Vancouver, Canada. PMLR 267, 2025. Copyright 2025 by the author(s).

Classical analyses of GD require the stepsizes to be sufficiently small so that the risk decreases monotonically (Nesterov, 2018). This is often referred to as the *descent lemma*. Specifically, by the definition of GD and the midpoint theorem, there exists $\mathbf{v}$ in between $\mathbf{w}_t$ and $\mathbf{w}_{t+1}$ such that

$$\mathcal{L}(\mathbf{w}_{t+1}) = \mathcal{L}(\mathbf{w}_t) - \eta_t \|\nabla \mathcal{L}(\mathbf{w}_t)\|^2$$
$$+ \frac{\eta_t^2}{2} \nabla \mathcal{L}(\mathbf{w}_t)^\top \nabla^2 \mathcal{L}(\mathbf{v}) \nabla \mathcal{L}(\mathbf{w}_t).$$

Thus a small stepsize $\eta_t < 2/\|\nabla^2 \mathcal{L}(\mathbf{v})\|$ guarantees the descent lemma. In this regime, a large volume of theory has been developed to show the convergence of GD in a variety of settings (see Lan, 2020, for example). We call this the *stable* regime.

However, in practice, deep learning models trained by GD often converge in the long run while suffering from a locally oscillatory risk (Wu et al., 2018; Xing et al., 2018; Lewkowycz et al., 2020; Cohen et al., 2021). This oscillation occurs when the stepsizes for GD are too large and the descent lemma is violated. This unstable convergence phenomenon is referred to by Cohen et al. (2021) as the *edge of stability* (EoS). Moreover, to obtain a reasonable optimization and generalization performance in deep learning practice, GD usually needs to operate in the EoS regime, instead of remaining in the stable regime (Wu et al., 2018; Cohen et al., 2021).

Recently, an interesting line of theoretical works showed the benefits of EoS for accelerating the convergence of GD (see for examples Altschuler & Parrilo, 2024a; Wu et al., 2024, other related works will be discussed later in Section 6). Specifically, Altschuler & Parrilo (2024a) proposed a *stepsize scheduler* for GD, in which GD occasionally violates the descent lemma but achieves a faster convergence rate for convex and smooth problems. The work by Wu et al. (2024) focused on GD with a *constant stepsize* for logistic regression with linearly separable data. They showed that a large constant stepsize leads to EoS but also a faster convergence. Note that the stepsizes considered in (Altschuler & Parrilo, 2024a; Wu et al., 2024) are *oblivious*, which are determined before the GD run and do not adapt to the evolving risk.

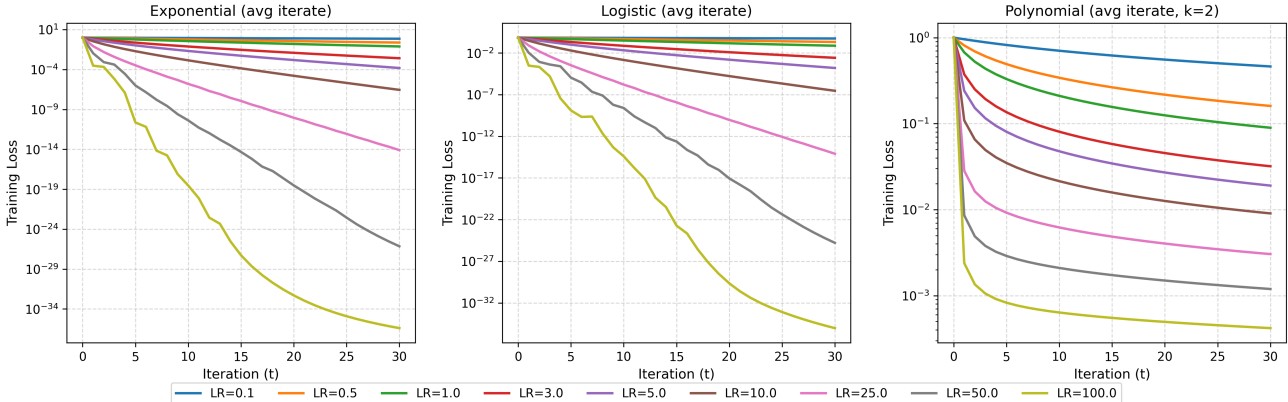

*Figure 1.* Simulation Results. We run gradient descent with stepsize scheduler (2) on exponential loss (left), logistic loss (middle), and polynomial loss (right) with linearly separable data. Here, we set $n = 100, \gamma = 0.25, d = 5$. We fix $\mathbf{w}^* \in \mathbb{R}^d, \|\mathbf{w}^*\|_2 = 1$ and sample $\mathbf{x}_i \in \mathbb{R}^d$ independently and uniformly from the unit sphere and accept it if $\langle \mathbf{x}_i, \mathbf{w}^* \rangle \geq \gamma$. Otherwise, we reject the data point. We keep sampling until we have $n = 100$ data. The y-axis is in log scale for all figures.

**Our results.** This work complements the prior theory by considering the convergence of GD with *adaptive stepsizes* in the EoS regime. Specifically, we consider logistic regression with linearly separable data, that is,

$$\mathcal{L}(\mathbf{w}) := \frac{1}{n} \sum_{i=1}^{n} \ell(y_i \mathbf{x}_i^\top \mathbf{w}), \quad \ell \in \{\ell_{\exp}, \ell_{\log}\}. \quad (1)$$

Here, the loss function can be the exponential loss or the logistic loss,

$$\ell_{\exp}(z) := \exp(-z), \quad \ell_{\log}(z) := \ln(1 + \exp(-z)),$$

and the dataset $(\mathbf{x}_i, y_i)_{i=1}^n$ is linearly separable, formalized by the following assumption.

**Assumption 1.1** (Linear separability). Assume that dataset $(\mathbf{x}_i, y_i)_{i=1}^n$ satisfies the following.

A. Assume, without loss of generality, that $\|\mathbf{x}_i\|_2 \leq 1$ and $y_i = 1$ for $i = 1, \ldots, n$.

B. Assume there exists $\gamma > 0$ and a unit vector $\mathbf{w}^*$ such that $y_i \mathbf{x}_i^\top \mathbf{w}^* \geq \gamma$ for $i = 1, \ldots, n$.

In this problem, it is known that the curvature becomes flatter as the risk decays. To compensate for this effect, we consider an adaptive stepsize scheduler (Ji & Telgarsky, 2021; Nacson et al., 2019; Wang et al., 2023a),

$$\eta_t := \eta \cdot (-\ell^{-1})' \circ \mathcal{L}(\mathbf{w}_t) \approx \eta / \mathcal{L}(\mathbf{w}_t),$$

where $\eta > 0$ is a fixed base stepsize. We make the following significant contributions.

**Benefits of large and adaptive stepsizes.** When $\eta$ is small, GD stays in the stable regime and achieves a convergence rate of $\exp(-\Theta(t))$, where $t$ is the number of steps (Ji & Telgarsky, 2021) (see also Proposition 2.1 in Section 2). However, we show GD can achieve an *arbitrarily fast* convergence rate by entering the EoS regime. Specifically, we show that for every $t \geq t_0$, the average of the first $t$ GD iterates achieves a risk upper bound of $\exp(-\Theta(\eta t))$ for *every* $\eta$. Here, $t_0$ is a function of the data margin but is independent of $\eta$. Therefore, one can use an arbitrarily large $\eta$ to obtain an arbitrarily small risk when $t \geq t_0$. By doing so, however, GD may enter the EoS regime.

**Lower bounds.** We then establish two lower bounds to complement our upper bounds. First, we provide an example of logistic regression with linearly separable data, where adaptive stepsize GD suffers from a risk lower bound of $\Theta(\exp(-t))$ if it does not enter the EoS regime. This, together with our upper bounds, demonstrates the benefits of EoS for accelerating the convergence of adaptive stepsize GD. Furthermore, we construct a hard example, in which every first-order gradient-based method must run for a margin-dependent number of steps to attain a small risk. This demonstrates the margin-dependent number of burn-in steps in our upper bound cannot be avoided in general.

**General losses and two-layer networks.** We also extend our results to a general class of loss functions, including polynomial loss (Ji & Telgarsky, 2021) and probit negative log-likelihood (Neal, 1997; Chib & Greenberg, 1998; Albert & Chib, 1993; Liu, 2004). Additionally, we show the same results hold for a two-layer network with leaky ReLU activation (Brutzkus et al., 2018). Notably, our analysis can be easily adapted to other common leaky activation functions

(Cai et al., 2024).

**Notation.** We use lowercase bold letters to denote vectors and $\|\cdot\|_2$ to denote the Euclidean $\ell_2$ norm for vectors. For positive integer $n$, we write $[n] := \{1, 2, ..., n\}$. Let $\mathbf{0}_d$ denote the zero vector in $\mathbb{R}^d$. We write $f(t) = \mathcal{O}(g(t))$ to mean that there exists a universal constant $c > 0$ such that $|f(t)| \leq c \cdot |g(t)|$ for sufficiently large $t$. Likewise, $f(t) = \Theta(g(t))$ means there exists a universal constant $c_1, c_2 > 0$ and $t_0$ such that $c_1 \cdot |g(t)| \leq |f(t)| \leq c_2 \cdot |g(t)|$ for $t \geq t_0$. We also use $\widetilde{\mathcal{O}}$ and $\widetilde{\Theta}$ to suppress constant and polylogarithmic factors. We let $C$ or $c$ denote universal constants, whose exact values may vary from line to line. $\mathrm{span}(\cdot)$ denotes the subspace spaned by a set of vectors. For $\mathbf{a} \in \mathbb{R}^d$ and $B \subset \mathbb{R}^d$, the Minkowski sum is defined as $\mathbf{a} + B := \{\mathbf{a} + \mathbf{b} : \mathbf{b} \in B\}$.

## 2. Logistic Regression

In this section, we present our improved analysis for GD with large and adaptive stepsizes for logistic regression with linearly separable data.

**Adaptive stepsizes.** We first explain the benefits of using adaptive stepsizes over a constant stepsize. Note that for logistic regression, the local sharpness $\|\nabla \mathcal{L}(\mathbf{w})\|$ is controlled by the risk $\mathcal{L}(\mathbf{w})$. Therefore as the risk decreases to zero (note that the dataset is separable), the local curvature becomes flatter. As a consequence, GD with a constant stepsize is less as less effective in the later stage. This observation has been exploited to show large stepsize GD enters a stable phase by Wu et al. (2024).

This issue can be mitigated by using the following adaptive stepsizes,

$$\eta_t = \eta \left(-\ell^{-1}\right)' \circ \mathcal{L}(\mathbf{w}_t) = \begin{cases} \dfrac{\eta}{\mathcal{L}(\mathbf{w}_t)} & \ell = \ell_{\exp}, \\ \dfrac{\eta \exp(\mathcal{L}(\mathbf{w}_t))}{\exp(\mathcal{L}(\mathbf{w}_t)) - 1} & \ell = \ell_{\log}. \end{cases} \quad (2)$$

When the risk $\mathcal{L}(\mathbf{w})$ becomes smaller, the stepsize becomes large to compensate for the flattened curvature. In this way, GD with adaptive stepsizes achieves a fast convergence rate $\exp(-\Theta(t))$ compared to that of GD with a constant stepsize, $\Theta(1/t)$, when both are in the stable regime (Nacson et al., 2019; Ji & Telgarsky, 2021; Wu et al., 2024).

Alternatively, (GD) with adaptive stepsizes (2) can be equivalently viewed as

$$\mathbf{w}_{t+1} = \mathbf{w}_t - \eta \nabla \phi(\mathbf{w}_t), \quad \phi(\mathbf{w}) := -\ell^{-1}(\mathcal{L}(\mathbf{w})), \quad (3)$$

that is, a constant-stepsize GD for a modified loss $\phi(\mathbf{w})$. This viewpoint is from (Ji & Telgarsky, 2021), where they utilized this idea to establish a primal-dual analysis of GD, obtaining an improved margin maximization rate.

**Prior analyses.** All the prior analyses for this adaptive stepsize GD relies on the descent lemma (Nacson et al., 2019; Ji & Telgarsky, 2021), hence they require the base stepsize $\eta > 0$ be small so that the risk decreases monotonically. The following proposition, a consequence of the main result in (Ji & Telgarsky, 2021), characterizes the best rate they can obtain in this regime.

**Proposition 2.1** (Consequences of (Ji & Telgarsky, 2021)). *Consider* (GD) *with adaptive stepsizes* (2) *for logistic regression* (1). *Suppose that Assumption 1.1 holds. Then $\phi$ is $\beta$-smooth with respect to $\ell_\infty$-norm, where $\beta = 1$ for exponential loss and $\beta \leq n$ for logistic loss. Moreover, for every $\eta \leq 1/\beta$, the risk $\mathcal{L}(\mathbf{w}_t)$ decreases monotonically and satisfies*

$$\mathcal{L}(\mathbf{w}_t) \leq C \exp\left(-\gamma^2 \eta t\right)$$

*where $C > 1$ is a universal constant.*

**An improved convergence rate.** Our first main result is an improved analysis of GD with adaptive stepsizes for logistic regression with linearly separable data, in which we allow GD to enter the EoS regime using a large stepsize. This is presented as the following theorem.

**Theorem 2.2** (An improved convergence rate). *Suppose that Assumption 1.1 holds. Consider* (GD) *with adaptive stepsizes* (2) *for logistic regression* (1). *Let $\overline{\mathbf{w}}_t := \frac{1}{t} \sum_{k=0}^{t-1} \mathbf{w}_k$ be the averaged iterates. Then for* every *$\eta > 0$, we have*

$$\mathcal{L}(\overline{\mathbf{w}}_t) \leq \begin{cases} \exp\left(-\dfrac{1}{4}\gamma^2 \eta t + \dfrac{\eta}{4\gamma^2 t}\right) & \ell = \ell_{\exp}, \\ \exp\left(-\dfrac{1}{4}\gamma^2 \eta t + \dfrac{\eta}{\gamma^2 t}\right) & \ell = \ell_{\log}. \end{cases}$$

*In particular, let the number of burn-in steps be*

$$t_0 := \begin{cases} \sqrt{2}/\gamma^2 & \ell = \ell_{\exp}, \\ 2\sqrt{2}/\gamma^2 & \ell = \ell_{\log}, \end{cases}$$

*then for every $t \geq t_0$, we have*

$$\mathcal{L}(\overline{\mathbf{w}}_t) \leq \exp\left(-\dfrac{1}{8}\gamma^2 \eta t\right),$$

*where the base stepsize $\eta$ can be arbitrarily large.*

Our Theorem 2.2 improves Proposition 2.1 by allowing the base stepsize $\eta$ to be arbitrarily large, where GD may not stay in the stable regime and is allowed to enter the EoS regime. More surprisingly, Theorem 2.2 implies that as soon as $t \geq t_0 = \Theta(1/\gamma^2)$, we have

$$\lim_{\eta \to \infty} \mathcal{L}(\overline{\mathbf{w}}_t) = 0.$$

This means GD with adaptive and large stepsizes converges *arbitrarily fast* for logistic regression with linearly separable data.

In sharp contrast to our result, for the same problem, GD with a constant stepsize can only achieve a $\mathcal{O}(1/t^2)$ rate even when operating in the EoS regime (Wu et al., 2024), and GD with adaptive stepsizes can only achieve a $\exp(-\Theta(t))$ rate when operating in the stable regime (see Proposition 2.1). Therefore, the improved rate fundamentally arises from combining both large and adaptive stepsizes.

It is also worth pointing out that GD with adaptive stepsizes can converge with an arbitrarily large based stepsize even under the exponential loss. In comparison, the work by Wu et al. (2023) provided an example where GD with a large constant stepsize cannot converge under the exponential loss, even when the dataset is linearly separable.

In the remaining part of this section, we provide the proof of Theorem 2.2.

*Proof of Theorem 2.2.* The proof of Theorem 5.2 uses the convexity of $\phi(\cdot)$ and a split optimization technique developed by Wu et al. (2024). Consider a comparator $\mathbf{u} = \mathbf{u}_1 + \mathbf{u}_2 \in \mathbb{R}^d$, then by (3) we have

$$\|\mathbf{w}_{t+1} - \mathbf{u}\|_2^2$$
$$= \|\mathbf{w}_t - \mathbf{u}\|_2^2 + 2\eta \langle \nabla\phi(\mathbf{w}_t), \mathbf{u} - \mathbf{w}_t \rangle + \eta^2 \|\nabla\phi(\mathbf{w}_t)\|_2^2$$
$$= \|\mathbf{w}_t - \mathbf{u}\|_2^2 + 2\eta \langle \nabla\phi(\mathbf{w}_t), \mathbf{u}_1 - \mathbf{w}_t \rangle$$
$$\qquad + \eta \left[ 2 \langle \nabla\phi(\mathbf{w}_t), \mathbf{u}_2 \rangle + \eta \|\nabla\phi(\mathbf{w}_t)\|_2^2 \right]$$
$$\leq \|\mathbf{w}_t - \mathbf{u}\|_2^2 + 2\eta \langle \nabla\phi(\mathbf{w}_t), \mathbf{u}_1 - \mathbf{w}_t \rangle \qquad (4)$$
$$\leq \|\mathbf{w}_t - \mathbf{u}\|_2^2 + 2\eta \left( \phi(\mathbf{u}_1) - \phi(\mathbf{w}_t) \right), \qquad (5)$$

where (4) is by properly setting $\mathbf{u}_2$ according to the following Lemma 2.3 and (5) is from the convexity of $\phi(\cdot)$ (see Theorem 5.2 in (Ji & Telgarsky, 2021), also Lemma G.1 in Appendix G).

**Lemma 2.3.** *Let* $\mathbf{u}_2 := (\eta/(2\gamma))\mathbf{w}^*$ *for exponential loss and* $\mathbf{u}_2 := (\eta/\gamma)\mathbf{w}^*$ *for logistic loss. Under Assumption 1.1, we have*

$$2 \langle \nabla\phi(\mathbf{w}), \mathbf{u}_2 \rangle + \eta \|\nabla\phi(\mathbf{w})\|_2^2 \leq 0.$$

The proof Lemma 2.3 is deferred to Appendix A. Going back to the proof, by rearranging (5) and telescoping the sum, we obtain

$$\frac{\|\mathbf{w}_t - \mathbf{u}\|_2^2}{2\eta t} + \frac{1}{t} \sum_{k=0}^{t-1} \phi(\mathbf{w}_k) \leq \phi(\mathbf{u}_1) + \frac{\|\mathbf{u}\|_2^2}{2\eta t}. \quad (6)$$

Set $\mathbf{u}_1 := (\gamma\eta t/2)\mathbf{w}^*$, then we have $\phi(\mathbf{u}_1) \leq -\gamma \|\mathbf{u}_1\|_2$ using Assumption 1.1 and the definition of $\phi(\cdot)$. This im-

plies that

$$\frac{1}{t} \sum_{k=0}^{t-1} \phi(\mathbf{w}_k) \leq -\gamma \|\mathbf{u}_1\|_2 + \frac{\|\mathbf{u}_1 + \mathbf{u}_2\|_2^2}{2\eta t}$$

$$\leq \begin{cases} -\dfrac{1}{4}\gamma^2\eta t + \dfrac{\eta}{4\gamma^2 t} & \ell = \ell_{\exp}, \\[2mm] -\dfrac{1}{4}\gamma^2\eta t + \dfrac{\eta}{\gamma^2 t} & \ell = \ell_{\log}. \end{cases}$$

We complete the proof by applying the convexity of $\phi(\cdot)$ and using that $\mathcal{L}(\cdot) = \ell(-\phi(\cdot))$. □

## 3. Lower Bounds

In this section, we establish two lower bounds on the convergence rate of GD for logistic regression with linearly separable data.

**A lower bound for GD in the stable regime.** Our next theorem provides a lower bound on the convergence rate of adaptive stepsize GD that stays in the stable regime.

**Theorem 3.1** (A lower bound for GD in the stable regime). *Consider* (GD) *with adaptive stepsizes* (2) *for logistic regression* (1) *with the following dataset*

$$\mathbf{x}_1 = (\gamma, \sqrt{1-\gamma^2}), \quad \mathbf{x}_2 = (\gamma, -\sqrt{1-\gamma^2}), \quad y_1 = y_2 = 1,$$

*where* $0 < \gamma < 0.1$. *It is clear that this dataset satisfies Assumption 1.1. If the base stepsize* $\eta$ *is such that* (GD) *induces a monotonically decreasing risk, then we have*

$$\mathcal{L}(\overline{\mathbf{w}}_t), \mathcal{L}(\mathbf{w}_t) \geq \exp(-ct), \quad t \geq 1,$$

*where* $c > 0$ *is a quantity depending on* $\gamma$ *but is independent of* $t$ *and* $\eta$.

Theorem 3.1 suggests that for adaptive stepsize GD that stays in the stable phase, the $\exp(-\Theta(t))$ rate from Proposition 2.1 is tight. Theorem 3.1 also demonstrates that to get the arbitrarily fast rate in Theorem 2.2, entering the EoS regime is unavoidable for adaptive stepsize GD.

The proof of Theorem 3.1 is motivated by a lower bound for constant-stepsize GD that stays in the stable phase in (Wu et al., 2024). We sketch the proof next. The proof is deferred to Appendix B.

*Proof sketch of Theorem 3.1.* One can show that on the constructed dataset, if $\mathcal{L}(\mathbf{w}_1) \leq \mathcal{L}(\mathbf{w}_0)$, then the base stepsize $\eta$ must be upper bounded by some constant $C_1$ which depends on $\ell(\cdot)$. Moreover, the split optimization bound implies an upper bound for both $\|\mathbf{w}_t\|_2$ and $\|\overline{\mathbf{w}}_t\|_2$ of order $\Theta(\eta t)$. Combining these two parts shows

$$\mathcal{L}(\overline{\mathbf{w}}_t) = \frac{1}{n} \sum_{i=1}^{n} \ell\left( \mathbf{x}_i^\top \overline{\mathbf{w}}_t \right) \geq \ell\left( \|\overline{\mathbf{w}}_t\|_2 \right) \geq \ell(ct)$$

for some constant $c$ that does not depend on $t$ or $\gamma$. The lower bound for $\mathcal{L}(\mathbf{w}_t)$ holds analogously. $\qquad\square$

**A lower bound for the number of burn-in steps.** We next provide a lower bound to show a certain number of burn-in steps are necessary for any first-order method.

**Theorem 3.2** (A lower bound for the number of burn-in steps)**.** *Let $\ell(\cdot) \geq 0$ be right-differentiable. Let $(\mathbf{w}_t)_{t\geq0}$ be given by a first-order method with $\mathbf{w}_0 := \mathbf{0}$, that is,*

$$\mathbf{w}_{t+1} \in \mathbf{w}_t + \mathsf{span}\left\{\nabla\mathcal{L}(\mathbf{w}_0), \dots, \nabla\mathcal{L}(\mathbf{w}_t)\right\}, \quad t \geq 0,$$

*where* $\mathsf{span}$ *is the linear span of a vector set, and $+$ is the Minkowski sum. Here, if $\ell(\cdot)$ is non-differentiable at some point, we use its right derivative at that point. Then there exists universal constants $c_1, c_2 > 0$ such that for every $0 < \gamma < c_1$, there exists a dataset $(\mathbf{x}_i, y_i)_{i=1}^n$ satisfying Assumption 1.1, in which the following holds. If*

$$\mathcal{L}(\mathbf{w}_t) \leq \ell(0)/2 \quad or \quad \mathcal{L}(\overline{\mathbf{w}}_t) \leq \ell(0)/2,$$

*for some $t \geq 0$, then*

$$t \geq c_2\gamma^{-2/3}.$$

Recall that in Theorem 2.2, adaptive stepsize GD needs a $\Theta(1/\gamma^2)$ number of burn-in steps to obtain the arbitrarily fast convergence rate. As a complement, our Theorem 3.2 suggests that an $\Omega(\gamma^{-2/3})$ number of burn-in steps are necessary for *every* first-order method.

Note that our Theorem 3.2 applies to a general loss function $\ell$, which does not have to be convex or differentiable. In particular, Theorem 3.2 can be applied to the Perceptron loss $\ell(z) := \max\{0, -z\}$, implying a lower bound on the number of steps for the Perceptron algorithm to converge (Novikoff, 1962). Although Perceptron is known to converge in at most $1/\gamma^2$ steps (Novikoff, 1962), our lower bound is the first of its kind to the best of our knowledge.

Although our Theorem 3.2 suggests a $\gamma$-dependent number of burn-in steps is unavoidable, we believe our lower bound is not tight. Specifically, we conjecture our lower bound can be improved to $\Omega(\gamma^{-1})$. This is left for future investigation.

The proof of Theorem 3.2 is motivated by the classical lower bound construction for first-order methods for convex and smooth optimization (Nesterov, 2018). We sketch its proof below. A full proof is deferred to Appendix C.

*Proof sketch of Theorem 3.2.* We use a dimensionality argument (Nesterov, 2018).Specifically, we construct a high-dimensional dataset such that for each gradient query, any first-order method can only collect information in one direction However, to decrease the loss below a constant, the algorithm must acquire information from $\Theta(d)$ distinct,

orthogonal directions, where $d$ is the dimension of the parameter. Consequently, starting from zero initialization, any first-order optimization algorithm requires at least $\Theta(d)$ iterations to reduce the loss below $\ell(0)/2$. Finally, exploiting both the boundedness and linear separability (with a positive margin $\gamma$) of the dataset, we derive an upper bound on $\gamma$ in terms of $d$. This bound implies a corresponding lower bound on the burn-in phase before the loss can drop below a constant fraction of its initial value. $\qquad\square$

# 4. Two-Layer Networks

In this section, we extend our results from a linear model to a two-layer network. Specifically, we consider a two-layer network with leaky ReLU activation (Brutzkus et al., 2018) defined as

$$f(\mathbf{w}; \mathbf{x}) := \frac{1}{m}\sum_{i=1}^m a_j\sigma\left(\mathbf{x}^\top\mathbf{w}^{(j)}\right), \quad \mathbf{w}^{(j)} \in \mathbb{R}^d, j \in [m],$$
(7)

where $m$ is the number of neurons, $a_j \in \{\pm1\}$ are fixed parameters, $\mathbf{w} := \left(\mathbf{w}^{(1)}, \mathbf{w}^{(2)}, ..., \mathbf{w}^{(m)}\right)^\top \in \mathbb{R}^{m\times d}$ is the trainable parameter, and $\sigma(\cdot)$ is the leaky ReLU activation defined as

$$\sigma(z) := \max\{z, \alpha z\}, \quad \alpha \in (0, 1).$$

We assume $\sigma'(0) = 1$ for convenience. The objective function is then given by

$$\mathcal{L}(\mathbf{w}) := \frac{1}{n}\sum_{i=1}^n \ell\left(y_i f(\mathbf{w}; \mathbf{x}_i)\right),$$
(8)

where $\ell(\cdot)$ is exponential loss or logistic loss. Similarly, we consider (GD) with adaptive stepsizes (2). Recall that (3) also applies here.

Similarly to logistic regression, we provide a convergence rate of adaptive stepsize GD for training two-layer networks with linearly separable data in the following theorem.

**Theorem 4.1** (A convergence rate for networks)**.** *Suppose that Assumption 1.1 holds. Consider (GD) with adaptive stepsizes (2) for objective (8), where the loss function $\ell(\cdot)$ is exponential loss or logistic loss. Then for* every *base stepsize $\eta > 0$, we have*

$$\min_{k<t}\mathcal{L}(\mathbf{w}_k) \leq \begin{cases} \exp\left(-\frac{1}{4}\alpha^2\gamma^2\eta t + \frac{\eta}{4\gamma^2 t}\right) & \ell = \ell_{\mathsf{exp}}, \\ \exp\left(-\frac{1}{4}\alpha^2\gamma^2\eta t + \frac{\eta}{\gamma^2 t}\right) & \ell = \ell_{\mathsf{log}}. \end{cases}$$

*Let the number of burn-in steps be*

$$t_0 := \begin{cases} \sqrt{2}/(\gamma^2\alpha) & \ell = \ell_{\mathsf{exp}}, \\ 2\sqrt{2}/(\gamma^2\alpha) & \ell = \ell_{\mathsf{log}}. \end{cases}$$

*Then for any $t \geq t_0$, we have*

$$\min_{k<t} \mathcal{L}(\mathbf{w}_k) \leq \exp\left(-\frac{1}{8}\alpha^2\gamma^2\eta t\right). \quad (9)$$

Our result in Theorem 4.1 provides a convergence rate similar to that of Theorem 2.2. By employing large and adaptive step sizes, GD can enter the EoS phase, which allows for faster convergence. In particular, once $t \geq t_0 = \Theta\left(1/\gamma^2\right)$, we have

$$\lim_{\eta \to \infty} \min_{k<t} \mathcal{L}(\mathbf{w}_k) = 0.$$

Note that for two-layer networks, we do not guarantee a corresponding loss upper bound for the averaged weights because $\phi(\cdot)$ need not to be convex. Instead, our guarantee holds for the *best* network along the GD trajectory, whose loss converges exponentially fast.

The work by Cai et al. (2024) provided a convergence rate for GD with constant stepsize for training two-layer networks with linearly separable data. They only obtained $\mathcal{O}(1/t^2)$ rate even if the constant-stepsize GD enters the EoS regime. By contrast, as shown in Theorem 4.1, adaptive stepsize GD can obtain an arbitrarily fast convergence.

Note that Theorem 4.1 can be extended to allow *leaky* variants of near-homogeneous activation functions, including leaky GeLU , leaky Softplus , and leaky SiLU (see Cai et al., 2024, for more examples). This is done in Appendix D.

The proof of Theorem 4.1 is motivated by (Wu et al., 2024; Cai et al., 2024) and is provided next.

*Proof of Theorem 4.1.* Consider a comparator $\mathbf{u} = \mathbf{u}_1 + \mathbf{u}_2 \in \mathbb{R}^{dm}$. Denote

$$\mathbf{u}_1 = \begin{pmatrix} \mathbf{u}_1^{(1)} \\ \mathbf{u}_1^{(2)} \\ \dots \\ \mathbf{u}_1^{(m)} \end{pmatrix}, \quad \mathbf{u}_2 = \begin{pmatrix} \mathbf{u}_2^{(1)} \\ \mathbf{u}_2^{(2)} \\ \dots \\ \mathbf{u}_2^{(m)} \end{pmatrix}.$$

From (3) we have

$$\|\mathbf{w}_{t+1} - \mathbf{u}\|_2^2$$
$$= \|\mathbf{w}_t - \mathbf{u}\|_2^2 + 2\eta m \underbrace{\langle \nabla\phi(\mathbf{w}_t), \mathbf{u}_1 - \mathbf{w}_t \rangle}_{I_1(\mathbf{w}_t)}$$
$$\eta \underbrace{\left(2\langle m \cdot \nabla\phi(\mathbf{w}_t), \mathbf{u}_2 \rangle + \eta \|m \cdot \nabla\phi(\mathbf{w}_t)\|_2^2\right)}_{I_2(\mathbf{w}_t)}. \quad (10)$$

We use the following two lemmas to complete the proof.

**Lemma 4.2.** *For $j \in [m]$, define $\mathbf{u}_2^{(j)} := \frac{a_j\eta}{2\gamma} \cdot \mathbf{w}^*$ for the exponential loss and $\mathbf{u}_2^{(j)} := \frac{a_j\eta}{\gamma} \cdot \mathbf{w}^*$ for the logistic loss.*

*Under Assumption 1.1, we have for every $\mathbf{w} \in \mathbb{R}^d$ that*

$$I_2(\mathbf{w}) := 2\langle m \cdot \nabla\phi(\mathbf{w}), \mathbf{u}_2 \rangle + \eta \|m \cdot \nabla\phi(\mathbf{w})\|_2^2 \leq 0.$$

**Lemma 4.3.** *For $j \in [m]$, define $\mathbf{u}_1^{(j)} := \frac{a_j}{2}\alpha\eta\gamma t \cdot \mathbf{w}^*$. Then, under Assumption 1.1, for any $\mathbf{w} \in \mathbb{R}^d$, we have*

$$I_1(\mathbf{w}) := \langle \nabla\phi(\mathbf{w}), \mathbf{u}_1 - \mathbf{w} \rangle \leq -\frac{\alpha\gamma}{m}\sum_{j=1}^{m}\left\|\mathbf{u}_1^{(j)}\right\|_2 - \phi(\mathbf{w}).$$

The proof of two lemmas above is deferred to Appendix D. Going back to the proof, we invoke the two lemmas above, rearrange (10), and telescope the sum, we get

$$\frac{\|\mathbf{w}_t - \mathbf{u}\|_2^2}{2\eta mt} + \frac{1}{t}\sum_{k=0}^{t-1}\phi(\mathbf{w}_k)$$
$$\leq -\frac{\alpha\gamma}{m}\sum_{j=1}^{m}\left\|\mathbf{u}_1^{(j)}\right\|_2 + \frac{\|\mathbf{u}\|_2^2}{2\eta mt}. \quad (11)$$

This implies

$$\frac{1}{t}\sum_{k=0}^{t-1}\phi(\mathbf{w}_k) \leq \begin{cases} -\frac{1}{4}\alpha^2\gamma^2\eta t + \frac{\eta}{4\gamma^2 t} & \ell = \ell_{\exp}, \\ -\frac{1}{4}\alpha^2\gamma^2\eta t + \frac{\eta}{\gamma^2 t} & \ell = \ell_{\log}. \end{cases}$$

We complete the proof by using the definition of $t_0$ in both cases and applying that $\mathcal{L}(\cdot) = \ell(-\phi(\cdot))$. $\quad\square$

## 5. A General Framework for GD Acceleration

In this section, we extend our results in Section 2 from exponential and logistic losses to a broad class of classification loss functions.

We begin by formulating general conditions for classification loss functions under which an accelerated convergence result like Theorem 2.2 can be established.

**Assumption 5.1** (Loss function conditions)**.** Let $\ell(\cdot) : \mathbb{R} \to \mathbb{R}$ be continuously differentiable. Assume that $\ell$ satisfies the following.

A. Assume that: The loss $\ell$ is positive, strictly decreasing, and convex. Its inverse $\ell^{-1}(z)$ exists and is differentiable for $z > 0$. Moreover, $\left(\ell^{-1}(z)\right)' < 0$ for $z > 0$.

B. Assume that $\psi(\cdot)$ defined as following is convex:

$$\psi(\mathbf{z}) := -\ell^{-1}\left(\frac{1}{n}\sum_{i=1}^{n}\ell(z_i)\right), \quad \mathbf{z} := (z_1, z_2, \dots, z_n). \quad (12)$$

C. Assume that: For any vector $\mathbf{z} = (z_1, z_2, \dots, z_n) \in \mathbb{R}^n$,

$$\frac{1}{n}\sum_{i=1}^{n}|\ell'(z_i)| \cdot \left|\left(-\ell^{-1}(z)\right)'\left(\frac{1}{n}\sum_{i=1}^{n}\ell(z_i)\right)\right| \leq C_\ell \quad (13)$$

for some constant $C_\ell > 0$.

Clearly, exponential and logistic losses satisfy Assumption 5.1A. They also satisfy Assumption 5.1B according to Theorem 5.1 in (Ji & Telgarsky, 2021) ( see also Lemma G.1). Moreover, they satisfy Assumption 5.1C with $C_\ell = 1$ for the exponential loss and $C_\ell = 2$ for the logistic loss (See Appendix A) Note that for $\phi$ in (3) we have

$$\phi(\mathbf{w}) = \psi\left((\mathbf{w}^\top \mathbf{x}_1, \mathbf{w}^\top \mathbf{x}_2, ..., \mathbf{w}^\top \mathbf{x}_n)^\top\right),$$

so $\phi(\cdot)$ is a composition between a linear mapping and $\psi(\cdot)$. Therefore, the convexity of $\psi(\cdot)$ directly implies the convexity of $\phi(\cdot)$. Condition (C) above indicates the self-boundedness of $\ell(\cdot)$, in the sense that the magnitude of $|\ell'(\cdot)|$ can be properly controlled by a function of $\ell(\cdot)$ itself, up to some constant multiplier.

Theorem 5.2 shows the acceleration brought by using large and adaptive stepsizes can happen for a broad class of classification losses. More surprisingly, Theorem 5.2 implies that for general losses, as soon as $t \geq t_0 = \Theta(1/\gamma^2)$, we also have

$$\lim_{\eta \to \infty} \mathcal{L}(\overline{\mathbf{w}}_t) = 0.$$

In particular, if $\ell(\cdot)$ has an exponential tail, i.e., $\ell(z) \lesssim \exp(-z)$ for large enough $z > 0$, then we can obtain a rate of $\mathcal{O}\left(\exp\left(-\Theta(\eta t)\right)\right)$ for large enough $t$.

**Improved convergence rate for general losses.** Now we present the acceleration for GD on the general classification losses that satisfy Assumption 5.1.

**Theorem 5.2** (An improved rate for general loss functions). *Suppose that Assumptions 1.1 and 5.1 hold. Consider* (GD) *with adaptive stepsizes* (2) *for objective function* $\mathcal{L}(\mathbf{w}) = \frac{1}{n}\sum_{i=1}^n \ell\left(y_i \mathbf{x}_i^\top \mathbf{w}\right)$. *Let* $\overline{\mathbf{w}}_t := \frac{1}{t}\sum_{k=0}^{t-1} \mathbf{w}_k$ *be the averaged iterates. Then for* every $\eta > 0$, *we have*

$$\mathcal{L}\left(\overline{\mathbf{w}}_t\right) \leq \ell\left(-\frac{1}{4}\gamma^2 \eta t + \frac{C_\ell \eta}{4\gamma^2 t}\right),$$

*where* $C_\ell$ *is the constant in Assumption 5.1. In particular, let the number of burn-in steps be*

$$t_0 := \sqrt{2}C_\ell/\gamma^2,$$

*then for every* $t \geq t_0$, *we have*

$$\mathcal{L}(\overline{\mathbf{w}}_t) \leq \ell\left(-\frac{1}{8}\gamma^2 \eta t\right),$$

*where the base stepsize* $\eta$ *can be arbitrarily large.*

Below we list two classification losses that satisfy Assumption 5.1 beyond exponential and logistic loss. The proof for these loss functions satisfying Assumption 5.1 is deferred to Appendix F.

1. **Polynomial loss.** For a fixed $k > 0$, we define (Ji & Telgarsky, 2021)

$$\ell(z) := \begin{cases} \dfrac{1}{(1+z)^k} & \text{for } z \geq 0 \\[2mm] -2kz + \dfrac{1}{(1-z)^k} & \text{for } z \leq 0 \end{cases}$$

One can show it satisfies Assumption 5.1C in Assumption 5.1 with $C_\ell = n^{1/k}$. Therefore, for any $t \geq \sqrt{2}n^{1/k}/\gamma^2$, it holds that

$$\mathcal{L}(\overline{\mathbf{w}}_t) \leq \left(1 + \frac{1}{8}\gamma^2 \eta t\right)^{-k}.$$

We show the simulation results for polynomial loss with $k = 2$ in Figure 1.

2. **Probit negative log-likelihood loss.** Let $F(z)$ denote the cumulative density function of one-dimensional standard Gaussian distribution. Then, we define (Albert & Chib, 1993; Neal, 1997; Chib & Greenberg, 1998; Liu, 2004)

$$\ell(z) := -\ln\left(F(z)\right).$$

It satisfies Assumption 5.1C with $C_\ell = 1$. Therefore, Theorem 5.2 implies that when $t \geq \sqrt{2}/\gamma^2$, it holds that

$$\begin{aligned} \mathcal{L}(\overline{\mathbf{w}}_t) &\leq -\ln\left(F\left(\frac{1}{8}\gamma^2 \eta t\right)\right) \\ &\leq -\ln\left(1 - \frac{c_1}{\gamma^2 \eta t} \cdot \exp\left(-c_2 \gamma^4 \eta^2 t^2\right)\right) \\ &\leq \mathcal{O}\left(\frac{1}{\gamma^2 \eta t} \cdot \exp\left(-c_2 \gamma^4 \eta^2 t^2\right)\right) \end{aligned}$$

for universal constants $c_1$ and $c_2$. The last inequality comes from classical Mill's ratio (See Lemma G.4).

## 6. Related Works

In this section, we discuss related papers.

**Edge of stability.** A growing body of theoretical work investigates *edge of stability* (EoS) under various scenarios. This includes quadratic functions (Zhu et al., 2022), certain non-convex functions (Wang et al., 2023b), single-neuron linear networks (Ahn et al., 2023), scalar neural networks (Kreisler et al., 2023), two-layer neural networks (Chen et al., 2023), scaled-Invariant networks (Lyu et al., 2022), diagonal linear networks (Even et al., 2023; Andriushchenko et al., 2023), and matrix factorization (Wang et al., 2021; Chen & Bruna, 2023). Several general frameworks have also been proposed to explain the EoS behavior, though they often rely on delicate assumptions (Kong & Tao, 2020; Ahn et al., 2022; Damian et al., 2022; Ma et al., 2022; Wang et al., 2022; Lu et al., 2023). In comparison, we focus on studying the benefits of EoS for improving optimization efficiency in logistic regression.

**Aggresive stepsize scheduler.** A recent line of work considered GD with an aggressive stepsize scheduler (Malitsky & Mishchenko, 2019; Altschuler & Parrilo, 2024a;b; Grimmer, 2024; Zhang et al., 2024). As their stepsize scheduler violates the descent lemma occasionally, they obtained an improved rate for GD in convex and smooth optimization (assuming the minimizer is finite). Instead, we consider GD with adaptive stepsizes that depend on the current risk, with a focus on logistic regression with linearly separable data.

**Logistic regression.** There is a large body of literature on the convergence and implicit bias of GD for logistic regression with linearly separable data. The work by Soudry et al. (2018); Ji & Telgarsky (2018) showed that small stepsize GD converges to the maximum margin direction. Their results also imply a risk convergence rate of $\mathcal{O}(1/t)$ (Soudry et al., 2018). Their results are extended to GD with a large constant stepsize by (Wu et al., 2023). In comparison, we focus on the effect of adaptive stepsizes.

Regarding this, Nacson et al. (2019) obtained a risk convergence rate of $\widetilde{\mathcal{O}}\left(\exp(-\sqrt{t})\right)$ by using adaptive stepsizes, and this is improved to $\mathcal{O}(\exp(-t))$ by the results in (Ji & Telgarsky, 2021), also using adaptive stepsizes. However, the adaptive stepsizes considered in (Nacson et al., 2019; Ji & Telgarsky, 2021) are relatively small, in which GD stays in the stable regime. In comparison, we consider GD with large and adaptive stepsizes, where GD is allowed to enter the EoS regime.

The works by (Wu et al., 2024; Cai et al., 2024) are also related to ours, where they studied GD with large constant stepsize in logistic regression and in two-layer networks, respectively. Although their GD can enter the EoS regime, the best risk convergence rate they obtained is $\mathcal{O}(1/t^2)$. In comparison, we considered GD with large and adaptive stepsizes for both cases, and we obtained an arbitrarily fast convergence rate. Finally, the work by Tyurin (2024) showed the equivalence between logistic regression with large fixed stepsize and batched Perceptron algorithm as the stepsize grows unboundedly. Their results are related to, but are not directly comparable with, ours.

## 7. Conclusion

In this paper, we show that GD with large and adaptive stepsizes achieves an arbitrarily fast convergence rate for logistic regression with linearly separable data. This holds as long as the number of steps is larger than a fixed threshold depending on the data margin. We also show the above is impossible if adaptive stepsize GD induces a monotonically decreasing risk, thereby demonstrating the benefit of unstable convergence. Moreover, we provide a lower bound, in which to achieve a small risk, every first-order method has to pay a number of steps as a function of the data margin.

This demonstrates a burn-in phase is necessary. Finally, we extend our results from logistic regression to a large class of loss functions and two-layer networks.

## Impact Statement

This paper presents work whose goal is to advance the field of Machine Learning. There are many potential societal consequences of our work, none of which we feel must be specifically highlighted here.

## Acknowledgement

We thank Yuhang Cai, Hossein Mobahi, and Matus Telgarsky for their helpful comments. We thank Qiuyu Ren for his significant help in improving the minimax lower bound. We thank Guy Kornowski and Ohad Shamir for pointing out missing references on the optimality of Perceptron. We gratefully acknowledge the NSF's support of FODSI through grant DMS-2023505 and of the NSF and the Simons Foundation for the Collaboration on the Theoretical Foundations of Deep Learning through awards DMS-2031883 and #814639 and of the ONR through MURI award N000142112431.

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

## A. Omitted Proofs for Theorem 2.2 in Section 2

**Properties of Loss Functions**   First, let's consider the properties of the exponential and logistic loss functions:

$$\ell_{\exp}(t) := \exp(-t), \quad \ell_{\log}(t) := \ln\left(1 + \exp(-t)\right).$$

We have:

- For $\ell = \ell_{\exp}$ or $\ell = \ell_{\log}$, we have $\ell > 0, \ell' < 0, \ell'' > 0$. Moreover, we have $|\ell'| = -\ell' \le \ell$, and the equation holds for exponential loss.

- For $\ell = \ell_{\log}$, $|\ell'(t)| \le 1$ for any $t \in \mathbb{R}$. This does not hold for the exponential loss.

- For $\ell = \ell_{\exp}$ or $\ell = \ell_{\log}$, the inverse function $\ell^{-1}(\cdot)$ exists and is continuously differentiable, and $\left(\ell^{-1}(t)\right)' < 0$ for $t > 0$.

- For $\ell = \ell_{\exp}$ or $\ell = \ell_{\log}$, we define $\psi(\mathbf{z}) := \left(-\ell^{-1}\right)'\left(\frac{1}{n}\sum_{i=1}^{n}\ell(z_i)\right)$ is convex for $\mathbf{z} = (z_1, z_2, ..., z_n)^\top$, so $\phi(\cdot)$ defined in (3) is also convex.

The first three arguments are easy to check, while the last one is a direct corollary of Theorem 5.1 in (Ji & Telgarsky, 2021).

*Proof of Lemma 2.3.*  When $\ell = \ell_{\exp}$, we have $\phi(\mathbf{w}) = \ln \mathcal{L}(\mathbf{w})$. Plug in $\mathbf{u}_2$, we have

$$2\left\langle \nabla \ln \mathcal{L}(\mathbf{w}), \mathbf{u}_2\right\rangle + \eta\left\|\nabla \ln \mathcal{L}(\mathbf{w})\right\|_2^2 = 2 \cdot \frac{\sum_{i=1}^{n}\ell_{\exp}'(\mathbf{w}^\top\mathbf{x}_i)\left\langle\mathbf{x}_i, \mathbf{u}_2\right\rangle}{\sum_{i=1}^{n}\ell_{\exp}(\mathbf{w}^\top\mathbf{x}_i)} + \eta\left\|\frac{\sum_{i=1}^{n}\ell_{\exp}'(\mathbf{w}^\top\mathbf{x}_i)\mathbf{x}_i}{\sum_{i=1}^{n}\ell_{\exp}(\mathbf{w}^\top\mathbf{x}_i)}\right\|_2^2$$

$$\le 2\gamma\left\|\mathbf{u}_2\right\|_2 \cdot \frac{\sum_{i=1}^{n}\ell_{\exp}'(\mathbf{w}^\top\mathbf{x}_i)}{\sum_{i=1}^{n}\ell_{\exp}(\mathbf{w}^\top\mathbf{x}_i)} + \eta\left(\frac{\sum_{i=1}^{n}|\ell_{\exp}'(\mathbf{w}^\top\mathbf{x}_i)|}{\sum_{i=1}^{n}\ell_{\exp}(\mathbf{w}^\top\mathbf{x}_i)}\right)^2 \qquad (\left\langle\mathbf{w}^*, \mathbf{x}_i\right\rangle \ge \gamma \text{ and } \|\mathbf{x}_i\|_2 \le 1 \text{ for any } 1 \le i \le n)$$

$$= 0. \qquad\qquad (\ell_{\exp}'(\cdot) = -\ell_{\exp}(\cdot) \text{ and } \|\mathbf{u}_2\|_2 = \frac{\eta}{2\gamma})$$

When $\ell = \ell_{\log}$, we set $\mathbf{u}_2 = \frac{\eta}{\gamma}\mathbf{w}^*$. Then,

$$2\left\langle \nabla\phi(\mathbf{w}), \mathbf{u}_2\right\rangle + \eta\left\|\nabla\phi(\mathbf{w})\right\|_2^2$$

$$= -\frac{2}{n}\cdot\left(\ell^{-1}\right)'\left(\mathcal{L}(\mathbf{w})\right)\cdot\sum_{i=1}^{n}\ell'(\mathbf{w}^\top\mathbf{x}_i)\left\langle\mathbf{u}_2, \mathbf{x}_i\right\rangle + \eta\left\|-\left(\ell^{-1}\right)'\left(\mathcal{L}(\mathbf{w})\right)\cdot\frac{1}{n}\sum_{i=1}^{n}\ell'(\mathbf{w}^\top\mathbf{x}_i)\mathbf{x}_i\right\|_2^2$$

$$\le -\frac{2\gamma\left\|\mathbf{u}_2\right\|_2}{n}\cdot\left|\left(\ell^{-1}\right)'\left(\mathcal{L}(\mathbf{w})\right)\right|\cdot\sum_{i=1}^{n}|\ell'(\mathbf{w}^\top\mathbf{x}_i)| + \eta\left(\left|\left(\ell^{-1}\right)'\left(\mathcal{L}(\mathbf{w})\right)\right|\cdot\frac{1}{n}\sum_{i=1}^{n}|\ell'(\mathbf{w}^\top\mathbf{x}_i)|\right)^2$$

$$\qquad\qquad (\|\mathbf{x}_i\|_2 \le 1 \text{ and } \left\langle\mathbf{x}_i, \mathbf{w}^*\right\rangle \ge \gamma)$$

$$= \frac{1}{n}\sum_{i=1}^{n}|\ell'(\mathbf{w}^\top\mathbf{x}_i)|\cdot\left|\left(\ell^{-1}\right)'\left(\mathcal{L}(\mathbf{w})\right)\right|\cdot\left[-2\gamma\left\|\mathbf{u}_2\right\|_2 + \eta\cdot\underbrace{\frac{1}{n}\sum_{i=1}^{n}|\ell'(\mathbf{w}^\top\mathbf{x}_i)|\cdot\left|\left(\ell^{-1}\right)'\left(\mathcal{L}(\mathbf{w})\right)\right|}_{\text{I}}\right].$$

Now we focus on the term I. Consider two cases. If $\mathcal{L}(\mathbf{w}) \ge \ln 2$, then

$$\text{I} \le \left|\left(\ell^{-1}\right)'\left(\mathcal{L}(\mathbf{w})\right)\right| = \frac{\exp(\mathcal{L}(\mathbf{w}))}{\exp(\mathcal{L}(\mathbf{w})) - 1} \le 2 \qquad (|\ell'(t)| \le 1 \text{ and } \frac{\exp(t)}{\exp(t)-1} \text{ is monotonically decreasing at } [\ln 2, \infty).)$$

Otherwise, we have $0 < \mathcal{L}(\mathbf{w}) < \ln 2$. Since $|\ell'(t)| \le \ell(t)$, we have

$$\text{I} \le \mathcal{L}(\mathbf{w})\cdot\left|\left(\ell^{-1}\right)'\left(\mathcal{L}(\mathbf{w})\right)\right| = h(\mathcal{L}(\mathbf{w})), \quad \text{where } h(t) := \frac{t\exp(t)}{\exp(t) - 1}.$$

We know

$$h'(t) = \frac{\exp(t)\left[\exp(t) - t - 1\right]}{\left(\exp(t) - 1\right)^2} \geq 0, \quad \lim_{t \to 0^-} h(t) = 1. \tag{14}$$

So $h(\cdot)$ is monotonically increasing and hence, $I \leq h(\mathcal{L}(\mathbf{w})) \leq h(\ln 2) = \ln 2$. Therefore, we have $I \leq 2$ and

$$2 \langle \nabla\phi(\mathbf{w}), \mathbf{u}_2 \rangle + \eta \|\nabla\phi(\mathbf{w})\|_2^2 \leq \frac{1}{n} \sum_{i=1}^{n} |\ell'(\mathbf{w}^\top \mathbf{x}_i)| \cdot \left|\left(\ell^{-1}\right)'(\mathcal{L}(\mathbf{w}))\right| \cdot \left[-2\gamma \|\mathbf{u}_2\|_2 + 2\eta\right] \leq 0. \tag{15}$$

$\square$

## B. Omitted Proof for Theorem 3.1 in Section 3

Before proving Theorem 3.1, let's first delve into two technical lemmas. First, we have the following lemma, which shows that for some special datasets, the stepsize must be small if the loss is non-monotonical.

**Lemma B.1** (Stepsize has to be small if the loss drops monotonically). *Let $\mathbf{w}_0 = \mathbf{0}_d$ and $\bar{\mathbf{x}} := 1./n \cdot \sum_{i=1}^{n} \mathbf{x}_i$. Assume there exist constants $r > 0$ and $q \in (0, 1)$ such that*

$$\frac{\left|\{i \in [n] : \mathbf{x}_i^\top \bar{\mathbf{x}} < -r\}\right|}{n} \geq q.$$

*Assume we perform the gradient descent in* (GD). *If $\mathcal{L}(\mathbf{w}_1) \leq \mathcal{L}(\mathbf{w}_0)$, then we must have*

$$\eta \leq \frac{qr}{\ell(0)}. \tag{16}$$

*Proof of Lemma B.1.* Since $\mathbf{w}_0 = \mathbf{0}_d$, we know $\mathcal{L}(\mathbf{w}_0) = \ell(0)$ and $\nabla\mathcal{L}(\mathbf{w}_0) = \ell'(0) \cdot \bar{\mathbf{x}}$ and

$$\nabla\phi(\mathbf{w}_0) = \left(-\ell^{-1}\right)'(\ell(0)) \cdot \ell'(0) \cdot \bar{\mathbf{x}}, \quad \mathbf{w}_1 = -\eta \left(-\ell^{-1}\right)'(\ell(0)) \cdot \ell'(0) \cdot \bar{\mathbf{x}}.$$

We know $-\left(-\ell^{-1}\right)'(\ell(0)) \cdot \ell'(0) = 1$ for both exponential loss and logistic loss. So for both losses, we have $\mathbf{w}_1 = \eta\bar{\mathbf{x}}$. Therefore, if $\eta > \frac{qr}{\ell(0)}$, this implies

$$\mathcal{L}(\mathbf{w}_1) = \frac{1}{n} \sum_{i=1}^{n} \ell(\eta\mathbf{x}_i^\top \bar{\mathbf{x}}) \geq \frac{1}{n} \sum_{i=1}^{n} \ell(\eta\mathbf{x}_i^\top \bar{\mathbf{x}})\mathbb{I}\left(\mathbf{x}_i^\top \bar{\mathbf{x}} < -r\right) \geq \ell(-\eta r) \cdot \frac{1}{n} \sum_{i=1}^{n} \mathbb{I}\left(\mathbf{x}_i^\top \bar{\mathbf{x}} < -r\right) \geq \ell(-\eta r) \cdot q$$

$$\text{(From the assumption)}$$

$$\geq q \cdot \ln\left(1 + \exp(\eta r)\right) \qquad\qquad\qquad \text{(For both exponential loss and logistic loss)}$$

$$\geq \eta q r \geq \ell(0) = \mathcal{L}(\mathbf{w}_0). \qquad\qquad\qquad\qquad\qquad\qquad \text{(From our assumption)}$$

This is a contradiction with our assumption of the stable phase, i.e., $\mathcal{L}(\mathbf{w}_1) \leq \mathcal{L}(\mathbf{w}_0)$. Therefore, we conclude $\square$

Then, we have the following lemma, which controls the upper bound of $\|\mathbf{w}_t\|_2$ along the optimization path (GD).

**Lemma B.2** (Upper Bound of $\|\mathbf{w}_t\|_2$). *Consider the gradient descent in* (GD) *for exponential loss or logistic loss starting from $\mathbf{w} = \mathbf{0}_d$, suppose the Assumption 1.1 holds, then for any $t \geq 1$ and $\eta > 0$, one has*

$$\|\mathbf{w}_t\|_2 \leq \left(\gamma + \frac{6}{\gamma}\right) \cdot \eta t. \tag{17}$$

*Proof of Lemma B.2.* We start from the split optimization bound (6). For both losses, from the proof of Theorem 2.2, we know the right-hand side of (6) is upper bounded by $-\frac{1}{4}\gamma^2\eta t + \frac{\eta}{\gamma^2 t}$. We also notice that $\|\mathbf{x}_i\|_2 \leq 1$ implies that for any $\mathbf{w} \in \mathbb{R}^d$,

$$-\|\mathbf{w}\|_2 \leq \phi(\mathbf{w}) = \left(-\ell^{-1}\right)\left(\frac{1}{n} \sum_{i=1}^{n} \ell\left(\mathbf{x}_i^\top \mathbf{w}\right)\right) \leq \|\mathbf{w}\|_2.$$

Combining this fact with (6) gives

$$\frac{\|\mathbf{w}_t - \mathbf{u}\|_2^2}{2\eta t} \leq -\frac{1}{t}\sum_{k=0}^{t-1}\phi(\mathbf{w}_k) - \frac{1}{4}\gamma^2\eta t + \frac{\eta}{\gamma^2 t} \leq \frac{1}{t}\sum_{k=0}^{t-1}\|\mathbf{w}_k\|_2 + \frac{\eta}{\gamma^2 t}$$

Here, $\mathbf{u} = (1/2\gamma\eta t + \eta/2\gamma)\cdot\mathbf{w}^*$ for the exponential loss and $\mathbf{u} = (1/2\gamma\eta t + \eta/\gamma)\cdot\mathbf{w}^*$ for the logistic loss, where $\mathbf{w}^* \in \mathbb{R}^d$ is the unit vector defined in Assumption 1.1. This implies for either loss and $t \geq 1$,

$$\|\mathbf{w}_t\|_2 \leq \|\mathbf{w}_t - \mathbf{u}\|_2 + \|\mathbf{u}\|_2 \leq \sqrt{2\eta\sum_{k=0}^{t-1}\|\mathbf{w}_k\|_2 + \frac{2\eta^2}{\gamma^2}} + \frac{1}{2}\gamma\eta t + \frac{\eta}{\gamma} \leq \sqrt{2\eta\sum_{k=0}^{t-1}\|\mathbf{w}_k\|_2} + \frac{1}{2}\gamma\eta t + \frac{3\eta}{\gamma},$$

where the last inequality above uses the fact that $\sqrt{a+b} \leq \sqrt{a} + \sqrt{b}$ for any $a, b > 0$. Obviously, (17) holds for $t = 0$. If it holds for $0, 1, 2, ..., t-1$ with $t \geq 1$, then for $t$, one has

$$\|\mathbf{w}_t\|_2 \leq \left(\sqrt{\left(\gamma + \frac{6}{\gamma}\right)} + \frac{1}{2}\left(\gamma + \frac{6}{\gamma}\right)\right)\eta t \leq \left(\gamma + \frac{6}{\gamma}\right)\cdot\eta t.$$

So (17) also holds for $t$. Invoking a simple induction completes the proof. $\qquad\square$

Now we prove the Theorem 3.1 by combining Lemma B.1 and Lemma B.2.

*Proof of Theorem 3.1.* One can easily check that the dataset in Theorem 3.1 satisfies the assumptions in Lemma B.1 with $r = 0.1$ and $q = 0.5$. So Lemma B.1 implies $\eta \leq c_1$ where $c_1$ is a constant that depends on the loss function. Combining this upper bound for the stepsize, triangle inequality and Lemma B.2, we get $\|\overline{\mathbf{w}}_t\|_2 \leq c_2 \cdot t$, where $c_2$ is a constant that does not depend on $t$ or $\eta$ (but it can depend on $\gamma$). Therefore, invoking the assumption that $\|\mathbf{x}_i\|_2 \leq 1$, $1 \leq i \leq n$, we have

$$\mathcal{L}(\overline{\mathbf{w}}_t) = \frac{1}{n}\sum_{i=1}^{n}\ell(\overline{\mathbf{w}}_t^\top\mathbf{x}_i) \geq \ell(c_2 \cdot t) \simeq \exp(-c_2 \cdot t),$$

where the last equivalence holds when $t$ is large due to the exponential tails of both losses. The guarantee for $\mathcal{L}(\mathbf{w}_t)$ holds analogously. $\qquad\square$

## C. Omitted Proofs for Theorem 3.2 in Section 3

*Proof of Theorem 3.2.* Without loss of generality, we can assume the dimension $d = n - 1$ and $n \geq 4$. We define $\mathbf{e}_i$ as the unit basis in $\mathbb{R}^d$ where the $i$-th entry is 1 and others are zero. Then, we construct a hard dataset $\mathcal{D}$. We set $y_i = 1$ for $1 \leq i \leq n$ and

$$\mathbf{x}_1 = \mathbf{e}_1, \quad \mathbf{x}_2 = -\frac{\sqrt{2}}{2}\mathbf{e}_1 + \frac{\sqrt{2}}{2}\mathbf{e}_2, \quad ..... \quad \mathbf{x}_{n-1} = -\frac{\sqrt{2}}{2}\mathbf{e}_{n-2} + \frac{\sqrt{2}}{2}\mathbf{e}_{n-1}, \quad \mathbf{x}_n = -\frac{\sqrt{2}}{2}\mathbf{e}_{n-1}.$$

This dataset satisfies the first condition in the theorem. For each small enough $\gamma > 0$, we construct a problem instance as follows. We set $d$ such that $\gamma \leq \sqrt{\frac{3}{d(d+1)(2d+1)}}$ and $\mathbf{w}^* := \sum_{i=1}^{d} i\cdot\sqrt{2}\gamma\cdot\mathbf{e}_i$. Then, we know $\|\mathbf{w}^*\|_2 = \sqrt{2}\gamma\sqrt{\sum_{i=1}^{d} i^2} = 1$ and $\langle y_i\mathbf{x}_i, \mathbf{w}^*\rangle \geq \gamma$ for $1 \leq i \leq n$. We denote $\mathbf{w} = (\mathbf{w}^{(1)}, \mathbf{w}^{(2)}, ..., \mathbf{w}^{(d)}) \in \mathbb{R}^d$. The loss function on the constructed dataset is

$$\mathcal{L}(\mathbf{w}) = \frac{1}{n}\left[\ell\left(\mathbf{w}^{(1)}\right) + \sum_{i=1}^{n-2}\ell\left(\frac{\sqrt{2}}{2}\left(-\mathbf{w}^{(i)} + \mathbf{w}^{(i+1)}\right)\right) + \ell\left(-\frac{\sqrt{2}}{2}\mathbf{w}^{(n-1)}\right)\right].$$

This implies

$$\frac{\partial \mathcal{L}(\mathbf{w})}{\partial \mathbf{w}^{(i)}} = \begin{cases} \frac{1}{n} \cdot \left[ \left( \ell'\left( \mathbf{w}^{(1)} \right) \right) - \frac{\sqrt{2}}{2} \ell'\left( \frac{\sqrt{2}}{2} \left( -\mathbf{w}^{(1)} + \mathbf{w}^{(2)} \right) \right) \right], & i = 1 \\[2ex] \frac{\sqrt{2}}{2n} \cdot \left[ \ell'\left( \frac{\sqrt{2}}{2} \left( -\mathbf{w}^{(i-1)} + \mathbf{w}^{(i)} \right) \right) - \ell'\left( \frac{\sqrt{2}}{2} \left( -\mathbf{w}^{(i)} + \mathbf{w}^{(i+1)} \right) \right) \right], & 2 \le i \le n-2 \\[2ex] \frac{\sqrt{2}}{2n} \cdot \left[ \ell'\left( \frac{\sqrt{2}}{2} \left( -\mathbf{w}^{(n-2)} + \mathbf{w}^{(n-1)} \right) \right) - \ell'\left( -\frac{\sqrt{2}}{2} \mathbf{w}^{(n-1)} \right) \right], & i = n-1. \end{cases}$$

Since $\mathbf{w}_0 = \mathbf{0}_d$, we know at the initial time, only the derivative of $\mathcal{L}(\mathbf{w})$ to the first entry is non-zero. We define $\mathbb{R}_k^d$ as the subset in $\mathbb{R}^d$ such that the last $d - k$ entries are all zero for $0 \le k \le d$. Specially, we know $\mathbb{R}_0^d = \{\mathbf{0}_d\}$ and $\mathbb{R}_d^d = \mathbb{R}^d$. A simple induction argument indicates that for any first-order gradient optimization algorithm, it holds that

$$\mathbf{w}_t \in \mathbb{R}_t^d, \quad \forall \quad 0 \le t \le d.$$

For any $0 \le t \le n$, if $\mathbf{w} \in \mathbb{R}_t^d$, the the last $d - t$ entries of $\mathbf{w}$ are all zero, which implies

$$\mathcal{L}(\mathbf{w}) \ge \frac{d - t}{n} = \frac{n - 1 - t}{n}.$$

From the definition of $\gamma$, we know

$$\gamma = \sqrt{\frac{3}{d(d+1)(2d+1)}} = \sqrt{\frac{3}{(n-1)n(2n-1)}} \ge \sqrt{\frac{3}{2n^3}},$$

which implies $n \ge \left( \frac{2\gamma^2}{3} \right)^{-\frac{1}{3}}$. Therefore, if $t \le \frac{1}{2} \left( \frac{2\gamma^2}{3} \right)^{-\frac{1}{3}} - 1 \simeq \gamma^{-\frac{2}{3}}$, then for all $0 \le k \le t$, since $\mathbf{w}_k \in \mathbb{R}_k^d$, it holds that

$$\mathcal{L}(\mathbf{w}_k) \ge \frac{d - k}{n} = \frac{n - 1 - k}{n} \ge \frac{1}{2}.$$

Moreover, for the average iterate $\overline{\mathbf{w}}_t := \frac{1}{t} \sum_{k=0}^{t-1} \mathbf{w}_k$, since $\overline{\mathbf{w}}_t \in \mathbb{R}_{t-1}^d$, we also have the same lower bound guarantee. Therefore, we conclude. $\qquad\square$

# D. Omitted Proofs for Theorem 4.1 in Section 4

## D.1. Assumptions and Examples on the Activation Functions

**Assumptions.** First, we provide general assumptions on the activation functions.

**Assumption D.1** (Activation Function). In (7), let the activation function $\sigma(z) : \mathbb{R} \to \mathbb{R}$ be differentiable at $z \ne 0$ and right-differentiable at $z = 0$. With a little abuse of notation, let $\sigma'(z)$ be its derivative at $z \ne 0$ and the right derivative at $z = 0$. Moreover, we assume

- $\sigma'(z)$ is continuous for $z \ne 0$ and right-continuous for $z = 0$. There exists $0 < \alpha < 1$ such that $\sigma'(z) \in (\alpha, 1]$.

- For any $z \in \mathbb{R}$, it holds that $|\sigma(z) - \sigma' z| \le \kappa$ for some $\kappa > 0$.

**Examples of activation functions.** Assumption D.1 holds for a broad class of leaky activations. For instance, let $\sigma_*(\cdot)$ be one of the following:

- GeLU: $\sigma_{\mathsf{GeLU}}(x) := x \cdot F(x)$. Here, $F(x)$ is the cumulative density function of standard Gaussian distribution.

- Softplus: $\sigma_{\mathsf{Softplus}}(x) := \log\left(1 + \exp(x)\right).$

- SiLU: $\sigma_{\mathsf{SiLU}}(x) := \frac{x}{1+\exp(-x)}$

- ReLU: $\sigma_{\mathsf{ReLU}}(x) := \max\{x, 0\}$.

Then, we fix some constant $0.5 < c < 1$ and define

$$\sigma(x) = c \cdot x + \frac{1-c}{4} \cdot \sigma_*(x), \tag{18}$$

where $\sigma_*$ can be any activation function above. It is straightforward to check that such a "leaky" variant satisfies Assumption D.1 with a uniform $\alpha = 0.25, \kappa = 1$.

*Proof for the examples of activation functions.* Now we prove the leaky activation functions above satisfy Assumption D.1.

- **GeLU:** One can easily show that $\sigma_{\mathsf{GeLU}}(x) = xF(x)$ and $\sigma'_{\mathsf{GeLU}}(x) = F(x) + xf(x), \sigma''_{\mathsf{GeLU}}(x) = (2-x^2)f(x)$, where $F$ and $f$ are the cumulative density function and probability density function of standard Gaussian distribution, respectively. Since $f \geq 0$, we know the global maximum of $\sigma_{\mathsf{GeLU}}(\cdot)$ is taken at either $x \to -\infty$ or $x = 1/\sqrt{2}$, while the global minimum is taken at either $x \to \infty$ or $x = -1/\sqrt{2}$. One can easily show that $\lim_{x\to\infty} \sigma'_{\mathsf{GeLU}}(x) = 1, \lim_{x\to-\infty} \sigma'_{\mathsf{GeLU}}(x) = 0$, and $\sigma'_{\mathsf{GeLU}}(\sqrt{2}/2) < 1.2, \sigma'_{\mathsf{GeLU}}(-\sqrt{2}/2) > -0.2$. Therefore, $\sigma = cx + \frac{1-c}{4} \cdot \sigma_{\mathsf{GeLU}}(x)$ satisfies the first condition in Assumption D.1 with $\alpha = 0.25$. Moreover, since $|\sigma_{\mathsf{GeLU}}(x) - \sigma'_{\mathsf{GeLU}}(x) \cdot x| = x^2 f(x) \leq 0.3$, we know that it satisfies the second condition in Assumption D.1 with $\kappa = \frac{1-c}{4} \cdot 0.3 \leq 0.3$.

  **Softplus:** One can easily know $0 < \sigma'_{\mathsf{Softplus}}(x) = \frac{\exp(x)}{1+\exp(x)} < 1$, so the first condition in Assumption D.1 holds with $\alpha = 0.25$. Define $h(x) = \sigma_{\mathsf{Softplus}}(x) - x \cdot \sigma'_{\mathsf{Softplus}}(x) = \log(1+\exp(x)) - \frac{x \exp(x)}{1+\exp(x)}$, we have $h'(x) = \frac{-x \exp(x)}{(1+\exp(x))^2}$, so we have $h(x) \geq \lim_{x\to\pm\infty} h(x) = 0$ and $h(x) \leq h(0) = \ln(2) \leq 1$. So the second condition holds with $\kappa \leq \frac{1-c}{4} \leq 1$.

  **SiLU :** For Swish function with a fixed $\beta > 0$, one can easily show that $\sigma'_{\mathsf{Swish}}(x) = \frac{1+\exp(-\beta x)(1-\beta x)}{(1+\exp(-\beta x))^2} \in [0,1]$, since $0 \leq 1 + \exp(-\beta x)(1-\beta x) \leq 1 + \exp(-2\beta x)$. So the first condition in Assumption D.1 holds with $\alpha = 0.25$. For the second condition, one has $\sigma_{\mathsf{Swish}}(x) - x \cdot \sigma'_{\mathsf{Swish}}(x) = \frac{\beta x^2 \exp(-\beta x)}{(1+\exp(-\beta x))^2} \in [0, 1/(2\beta)]$. So the second condition holds with $\kappa = \frac{1-c}{4 \cdot 2\beta} \leq \frac{1}{2\beta}$.

  **ReLU:** Since $0 \leq \sigma'_{\mathsf{ReLU}}(x) \leq 1$, one has the first condition in Assumption D.1 holds with $\alpha = 0.5$. The second condition holds with $\kappa = 1$ trivially.

$\square$

## D.2. Omitted Proof of Theorem 4.1

Before proving Theorem 4.1, we first present the general versions of two lemmas in Section 4.

**Lemma D.2.** *We take* $\mathbf{u}_2^{(j)} = \frac{a_j C_\ell \eta}{2\gamma} \cdot \mathbf{w}^*$ *for* $j = 1, 2, ..., m$, *where* $C_\ell > 0$ *is the constant in Condition (C) in the Assumption 5.1.* $C_\ell = 1$ *for the exponential loss and* $C_\ell = 2$ *for the logistic loss. Then, for every* $\mathbf{w} \in \mathbb{R}^d$, *one has*

$$I_2(\mathbf{w}) := 2 \langle m \cdot \nabla \phi(\mathbf{w}), \mathbf{u}_2 \rangle + \eta \|m \cdot \nabla \phi(\mathbf{w})\|_2^2 \leq 0.$$

*Proof of Lemma D.2.* We define

$$g_{i,j} := \ell'\left(y_i f(\mathbf{w}, \mathbf{x}_i)\right) \cdot \sigma'\left(\mathbf{x}_i^\top \mathbf{w}^{(j)}\right) \leq 0,$$

then we have

$$I_2(\mathbf{w}) = \frac{2}{n}\left(-\ell^{-1}\right)'(\mathcal{L}(\mathbf{w})) \cdot \sum_{j=1}^m \sum_{i=1}^n g_{i,j} \cdot \left\|\mathbf{u}_2^{(j)}\right\|_2 \cdot y_i \mathbf{x}_i^\top \mathbf{w}^* + \eta \cdot \sum_{j=1}^m \left\|\frac{a_j}{n} \cdot \left(-\ell^{-1}\right)'(\mathcal{L}(\mathbf{w})) \cdot \sum_{i=1}^n g_{i,j} y_i \mathbf{x}_i\right\|_2^2.$$

Since $g_{i,j} \leq 0$ for any $1 \leq i \leq n, 1 \leq j \leq m$, and $y_i \mathbf{x}_i^\top \mathbf{w}^* \geq \gamma > 0, \|\mathbf{x}_i\|_2 \leq 1$, one has

$$I_2(\mathbf{w}) \leq -\frac{2\gamma}{n} \cdot \left(-\ell^{-1}\right)'(\mathcal{L}(\mathbf{w})) \cdot \sum_{j=1}^m \sum_{i=1}^n \left\|\mathbf{u}_2^{(j)}\right\|_2 \cdot |g_{i,j}| + \frac{\eta}{n^2}\left(-\ell^{-1}\right)'(\mathcal{L}(\mathbf{w}))^2 \sum_{j=1}^m \left(\sum_{i=1}^n |g_{i,j}|\right)^2$$

$$= \left(-\ell^{-1}\right)'(\mathcal{L}(\mathbf{w})) \cdot \frac{1}{n} \sum_{j=1}^m \sum_{i=1}^n |g_{i,j}| \cdot \left(-2\gamma \left\|\mathbf{u}_2^{(j)}\right\|_2 + \eta \cdot \left(-\ell^{-1}\right)'(\mathcal{L}(\mathbf{w})) \cdot \frac{1}{n} \sum_{k=1}^n |g_{k,j}|\right). \tag{19}$$

For any $1 \leq k \leq n$, from the Assumption 5.1, we have

$$
\begin{aligned}
\left(-\ell^{-1}\right)'\left(\mathcal{L}(\mathbf{w})\right) \cdot \frac{1}{n} \sum_{k=1}^{n} |g_{k,j}| &= \frac{1}{n} \sum_{k=1}^{n} \left(-\ell^{-1}\right)'\left(\mathcal{L}(\mathbf{w})\right) \cdot |\ell'\left(f\left(\mathbf{w}, \mathbf{x}_i\right)\right)| \cdot \sigma'\left(\mathbf{x}_i^\top \mathbf{w}^*\right) \\
&\leq \frac{1}{n} \sum_{k=1}^{n} \left(-\ell^{-1}\right)'\left(\mathcal{L}(\mathbf{w})\right) \cdot |\ell'\left(f\left(\mathbf{w}, \mathbf{x}_i\right)\right)| \qquad (\sigma'(\cdot) \leq 1) \\
&\leq C_\ell,
\end{aligned}
$$

where $C_\ell > 0$ is the constant in Assumption 5.1. Therefore, combining the above upper bound, equation (19), and the definition of $\mathbf{u}_2$, we conclude. $\qquad \square$

**Lemma D.3.** *We take $\mathbf{u}_1^{(j)} = a_j \cdot \left\|\mathbf{u}_1^{(j)}\right\|_2 \cdot \mathbf{w}^*$. Then, we have for any $\mathbf{w} \in \mathbb{R}^d$, it holds that*

$$
I_1(\mathbf{w}) := \langle \nabla \phi(\mathbf{w}), \mathbf{u}_1 - \mathbf{w} \rangle \leq \kappa - \frac{\alpha\gamma}{m} \sum_{j=1}^{m} \left\|\mathbf{u}_1^{(j)}\right\|_2 - \phi(\mathbf{w})
$$

*Proof of Lemma D.3.* From the definition of $\phi(\cdot)$, one has

$$
\begin{aligned}
I_1(\mathbf{w}) &:= \langle \nabla \phi(\mathbf{w}), \mathbf{u}_1 - \mathbf{w} \rangle \\
&= \left(-\ell^{-1}\right)'\left(\mathcal{L}(\mathbf{w})\right) \cdot \frac{1}{n} \sum_{i=1}^{n} \ell'\left(y_i f\left(\mathbf{w}; \mathbf{x}_i\right)\right) \cdot \frac{1}{m} \sum_{j=1}^{m} y_i a_j \sigma'\left(\mathbf{x}_i^\top \mathbf{w}^{(j)}\right) \cdot \mathbf{x}_i^\top \left(\mathbf{u}_1^{(j)} - \mathbf{w}^{(j)}\right) \\
&= \left(-\ell^{-1}\right)'\left(\mathcal{L}(\mathbf{w})\right) \cdot \frac{1}{n} \sum_{i=1}^{n} \ell'\left(y_i f\left(\mathbf{w}; \mathbf{x}_i\right)\right) \cdot \left[J_i - y_i f\left(\mathbf{w}, \mathbf{x}_i\right)\right],
\end{aligned}
$$

where

$$
\begin{aligned}
J_i &:= \frac{1}{m} \sum_{j=1}^{m} a_j \left[\sigma'\left(\mathbf{x}_i^\top \mathbf{w}^{(j)}\right) y_i \mathbf{x}_i^\top \mathbf{u}_1^{(j)}\right] + \frac{1}{m} \sum_{j=1}^{m} y_i a_j \underbrace{\left[\sigma\left(\mathbf{x}_i^\top \mathbf{w}^{(j)}\right) - \sigma'\left(\mathbf{x}_i^\top \mathbf{w}^{(j)}\right) \mathbf{x}_i^\top \mathbf{w}^{(j)}\right]}_{|\cdot| \leq \kappa} \\
&\geq \frac{1}{m} \sum_{j=1}^{m} a_j \left[\sigma'\left(\mathbf{x}_i^\top \mathbf{w}^{(j)}\right) y_i \mathbf{x}_i^\top \mathbf{u}_1^{(j)}\right] - \kappa = \frac{1}{m} \sum_{j=1}^{m} \left\|\mathbf{u}_1^{(j)}\right\|_2 \left[\underbrace{\sigma'\left(\mathbf{x}_i^\top \mathbf{w}^{(j)}\right)}_{\geq \alpha} \cdot \underbrace{y_i \mathbf{x}_i^\top \mathbf{w}^*}_{\geq \gamma}\right] - \kappa \\
&\geq \frac{\alpha\gamma}{m} \sum_{j=1}^{m} \left\|\mathbf{u}_1^{(j)}\right\|_2 - \kappa.
\end{aligned}
$$

Notice $\ell'(\cdot) \leq 0$, we define $\mathbf{z} := \left(y_1 f\left(\mathbf{w}; \mathbf{x}_1\right), y_2 f\left(\mathbf{w}, \mathbf{x}_2\right), ..., y_n f\left(\mathbf{w}, \mathbf{x}_n\right)\right)^\top$ and $\mathbf{1}_n = (1, 1, ..., 1)^\top \in \mathbb{R}^n$. Then, from the definition of $\psi(\cdot)$, one has

$$
\begin{aligned}
I_1(\mathbf{w}) &\leq \left(-\ell^{-1}\right)'\left(\mathcal{L}(\mathbf{w})\right) \cdot \frac{1}{n} \sum_{i=1}^{n} \ell'\left(y_i f\left(\mathbf{w}; \mathbf{x}_i\right)\right) \cdot \left(\frac{\alpha\gamma}{m} \sum_{j=1}^{m} \left\|\mathbf{u}_1^{(j)}\right\|_2 - \kappa - y_i f\left(\mathbf{w}, \mathbf{x}_i\right)\right) \\
&= \left\langle \nabla \psi(\mathbf{z}), \left(\frac{\alpha\gamma}{m} \sum_{j=1}^{m} \left\|\mathbf{u}_1^{(j)}\right\|_2 - \kappa\right) \cdot \mathbf{1}_n - \mathbf{z} \right\rangle \leq \psi\left(\left(\frac{\alpha\gamma}{m} \sum_{j=1}^{m} \left\|\mathbf{u}_1^{(j)}\right\|_2 - \kappa\right) \cdot \mathbf{1}_n\right) - \psi(\mathbf{z}) \\
&\qquad\qquad\qquad\qquad\qquad\qquad\qquad\qquad\qquad\qquad\qquad\qquad\qquad\qquad\qquad\qquad \text{(Since } \psi(\cdot) \text{ is convex)} \\
&= \kappa - \frac{\alpha\gamma}{m} \sum_{j=1}^{m} \left\|\mathbf{u}_1^{(j)}\right\|_2 - \phi(\mathbf{w}) \qquad\qquad\qquad\qquad\qquad \text{(From the definition of } \psi(\cdot) \text{ and } \phi(\cdot)\text{)}
\end{aligned}
$$

$\qquad\qquad\qquad\qquad\qquad\qquad\qquad\qquad\qquad\qquad\qquad\qquad\qquad\qquad\qquad\qquad\qquad\qquad\qquad\qquad\qquad\qquad\qquad\qquad\qquad\qquad \square$

Now we present the proof for Theorem 4.1.

*Proof of Theorem 4.1.* We consider a comparator $\mathbf{u} = \mathbf{u}_1 + \mathbf{u}_2$, where

$$
\mathbf{u}_1 = \begin{pmatrix} \mathbf{u}_1^{(1)} \\ \mathbf{u}_1^{(2)} \\ ... \\ \mathbf{u}_1^{(m)} \end{pmatrix}, \quad \mathbf{u}_2 = \begin{pmatrix} \mathbf{u}_2^{(1)} \\ \mathbf{u}_2^{(2)} \\ ... \\ \mathbf{u}_2^{(m)} \end{pmatrix}.
$$

Consider the following decomposition:

$$
\begin{aligned}
\|\mathbf{w}_{t+1} - \mathbf{u}\|_2^2 &= \|\mathbf{w}_t - \mathbf{u}\|_2^2 + 2m\eta_t \langle \nabla \mathcal{L}(\mathbf{w}_t), \mathbf{u} - \mathbf{w}_t \rangle + m^2 \eta_t^2 \|\nabla \mathcal{L}(\mathbf{w}_t)\|_2^2 \\
&= \|\mathbf{w}_t - \mathbf{u}\|_2^2 + 2\eta m \underbrace{\langle \nabla \phi(\mathbf{w}_t), \mathbf{u}_1 - \mathbf{w}_t \rangle}_{I_1(\mathbf{w}_t)} + \eta \underbrace{\left( 2 \langle m \cdot \nabla \phi(\mathbf{w}_t), \mathbf{u}_2 \rangle + \eta \|m \cdot \nabla \phi(\mathbf{w}_t)\|_2^2 \right)}_{I_2(\mathbf{w}_t)}.
\end{aligned}
$$

Following Lemma D.3 and Lemma D.2 and inserting $\mathbf{w}_0 = \mathbf{0}_d$, we get a split optimization bound:

$$
\frac{\|\mathbf{w}_t - \mathbf{u}\|_2^2}{2\eta m t} + \frac{1}{t} \sum_{k=0}^{t-1} \phi(\mathbf{w}_k) \leq \kappa - \frac{\alpha \gamma}{m} \sum_{j=1}^{m} \left\| \mathbf{u}_1^{(j)} \right\|_2 + \frac{\|\mathbf{u}_1 + \mathbf{u}_2\|_2^2}{2\eta m t}. \tag{20}
$$

This implies

$$
\frac{1}{t} \sum_{k=0}^{t-1} \phi(\mathbf{w}_k) \leq \left( \kappa + \sum_{j=1}^{m} \left( -\frac{\alpha \gamma}{m} \left\| \mathbf{u}_1^{(j)} \right\|_2 + \frac{\left\| \mathbf{u}_1^{(j)} \right\|_2^2}{\eta m t} \right) \right) + \frac{\|\mathbf{u}_2\|_2^2}{\eta m t}.
$$

Taking

$$
\mathbf{u}_1^{(j)} = \frac{1}{2} \alpha \gamma \eta t \cdot \mathbf{w}^*, \quad \mathbf{u}_2^{(j)} = \frac{a_j C_\ell \eta}{2\gamma} \cdot \mathbf{w}^*, \quad 1 \leq j \leq m,
$$

we get

$$
\frac{1}{t} \sum_{k=0}^{t-1} \phi(\mathbf{w}_k) \leq \kappa - \frac{1}{4} \alpha^2 \gamma^2 \eta t + \frac{C_\ell^2 \eta}{4\gamma^2 t}.
$$

Therefore, we complete the proof of Theorem 4.1 by invoking the convexity of $\psi(\cdot)$. $\qquad \square$

## E. Omitted Proof of Theorem 5.2 in Section 5

The entire proof follows the proof of Theorem 2.2. First, we have the following key lemma which is an analogue of Lemma 2.3

**Lemma E.1** (Key Lemma). *We define* $\mathbf{u}_2 = \frac{C_\ell \eta}{2\gamma} \cdot \mathbf{w}^*$ *for a loss function* $\ell$ *satisfying Assumption 5.1 with loss specific constant* $C_\ell > 0$. *Under the definition of* $\phi(\cdot)$, *for any* $\mathbf{w} \in \mathbb{R}^d$ *and any* $\eta > 0$, *it holds that*

$$
2 \langle \nabla \phi(\mathbf{w}), \mathbf{u}_2 \rangle + \eta \|\nabla \phi(\mathbf{w})\|_2^2 \leq 0. \tag{21}
$$

*Proof of Lemma E.1.* Note that

$$2 \langle \nabla\phi(\mathbf{w}), \mathbf{u}_2 \rangle + \eta \|\nabla\phi(\mathbf{w})\|_2^2$$

$$= -\frac{2}{n} \cdot \left(\ell^{-1}\right)' (\mathcal{L}(\mathbf{w})) \cdot \sum_{i=1}^n \ell'(\mathbf{w}^\top \mathbf{x}_i) \langle \mathbf{u}_2, \mathbf{x}_i \rangle + \eta \left\| -\left(\ell^{-1}\right)' (\mathcal{L}(\mathbf{w})) \cdot \frac{1}{n} \sum_{i=1}^n \ell'(\mathbf{w}^\top \mathbf{x}_i)\mathbf{x}_i \right\|_2^2$$

$$\leq -\frac{2\gamma \|\mathbf{u}_2\|_2}{n} \cdot \left|\left(\ell^{-1}\right)' (\mathcal{L}(\mathbf{w}))\right| \cdot \sum_{i=1}^n |\ell'(\mathbf{w}^\top \mathbf{x}_i)| + \eta \left( \left|\left(\ell^{-1}\right)' (\mathcal{L}(\mathbf{w}))\right| \cdot \frac{1}{n} \sum_{i=1}^n |\ell'(\mathbf{w}^\top \mathbf{x}_i)| \right)^2$$

$$\hspace{6cm} (\|\mathbf{x}_i\|_2 \leq 1 \text{ and } \langle \mathbf{x}_i, \mathbf{w}^* \rangle \geq \gamma)$$

$$= \frac{1}{n} \sum_{i=1}^n |\ell'(\mathbf{w}^\top \mathbf{x}_i)| \cdot \left|\left(\ell^{-1}\right)' (\mathcal{L}(\mathbf{w}))\right| \cdot \left[ -2\gamma \|\mathbf{u}_2\|_2 + \eta \cdot \underbrace{\frac{1}{n} \sum_{i=1}^n |\ell'(\mathbf{w}^\top \mathbf{x}_i)| \cdot \left|\left(\ell^{-1}\right)' (\mathcal{L}(\mathbf{w}))\right|}_{\text{I}} \right]$$

$$\leq \frac{1}{n} \sum_{i=1}^n |\ell'(\mathbf{w}^\top \mathbf{x}_i)| \cdot \left|\left(\ell^{-1}\right)' (\mathcal{L}(\mathbf{w}))\right| \cdot [-2\gamma \|\mathbf{u}_2\|_2 + \eta C_\ell]. \hspace{2cm} \text{(From Assumption D.1)}$$

Invoking the definition of $\mathbf{u}_2$ completes the proof. $\qquad\square$

Now we prove the Theorem 5.2.

*Proof of Theorem 5.2.* Denote $\mathbf{u} = \mathbf{u}_1 + \mathbf{u}_2$, where $\mathbf{u}_2 = \frac{C_\ell \eta}{2\gamma} \cdot \mathbf{w}^* \in \mathbb{R}^d$. Then, we have Recall the gradient descent iterate (GD), the learning rate scheduler (2), and the definition for $\phi(\cdot)$. Then, following the proof of Theorem 2.2, we have

$$\|\mathbf{w}_{t+1} - \mathbf{u}\|_2^2$$
$$= \|\mathbf{w}_t - \mathbf{u}\|_2^2 + 2\eta \langle \nabla\phi(\mathbf{w}_t), \mathbf{u} - \mathbf{w}_t \rangle + \eta^2 \|\nabla\phi(\mathbf{w}_t)\|_2^2$$
$$= \|\mathbf{w}_t - \mathbf{u}\|_2^2 + 2\eta \langle \nabla\phi(\mathbf{w}_t), \mathbf{u}_1 - \mathbf{w}_t \rangle + \eta \left[ 2 \langle \nabla\phi(\mathbf{w}_t), \mathbf{u}_2 \rangle + \eta \|\nabla\phi(\mathbf{w}_t)\|_2^2 \right]$$
$$\overset{(a)}{\leq} \|\mathbf{w}_t - \mathbf{u}\|_2^2 + 2\eta \langle \nabla\phi(\mathbf{w}_t), \mathbf{u}_1 - \mathbf{w}_t \rangle$$
$$\overset{(b)}{\leq} \|\mathbf{w}_t - \mathbf{u}\|_2^2 + 2\eta \left( \phi(\mathbf{u}_1) - \phi(\mathbf{w}_t) \right). \tag{22}$$

Here, $(a)$ is from Lemma E.1, and $(b)$ is from the convexity of $\phi(\cdot)$ from Condition (B) in Assumption 5.1. Telescoping (22) from $k = 0$ to $k = t-1$ yields

$$\frac{\|\mathbf{w}_t - \mathbf{u}\|_2^2}{2\eta t} + \frac{1}{t} \sum_{k=0}^{t-1} \phi(\mathbf{w}_k) \leq \phi(\mathbf{u}_1) + \frac{\|\mathbf{w}_0 - \mathbf{u}\|_2^2}{2\eta t}. \tag{23}$$

We take

$$\mathbf{u}_1 = \frac{1}{2}\gamma\eta t \cdot \mathbf{w}^*, \quad \mathbf{w}_0 = \mathbf{0}_d.$$

Recall $\langle \mathbf{w}^*, \mathbf{x}_i \rangle \geq \gamma$ and the definition of $\phi(\cdot)$, we have $\phi(\mathbf{u}_1) \leq -\gamma \|\mathbf{u}_1\|_2$. This implies

$$\frac{1}{t} \sum_{k=0}^{t-1} \phi(\mathbf{w}_k) \leq -\gamma \|\mathbf{u}_1\|_2 + \frac{\|\mathbf{u}_1 + \mathbf{u}_2\|_2^2}{2\eta t} \leq -\gamma \|\mathbf{u}_1\|_2 + \frac{\|\mathbf{u}_1\|_2^2 + \|\mathbf{u}_2\|_2^2}{\eta t} = -\frac{1}{4}\gamma^2 \eta t + \frac{C_\ell \eta}{4\gamma^2 t}.$$

Therefore, when $t \geq t_0 := \sqrt{2}C_\ell/\gamma^2$,

$$\frac{1}{t} \sum_{k=0}^{t-1} \phi(\mathbf{w}_k) \leq -\frac{1}{8}\gamma^2 \eta t$$

Invoking the convexity of $\phi(\cdot)$ completes the proof. $\qquad\square$

## F. Examples of Loss Functions

**Example: polynomial loss.** We consider the polynomial loss (Ji & Telgarsky, 2021). For a fixed $k > 0$, we define

$$\ell(z) := \begin{cases} \dfrac{1}{(1+z)^k} & \text{for } z \geq 0 \\ -2kz + \dfrac{1}{(1-z)^k} & \text{for } z \leq 0 \end{cases} \tag{24}$$

Then, we have

$$\ell'(z) := \begin{cases} \dfrac{-k}{(1+z)^{k+1}} & \text{for } z \geq 0 \\ -2k + \dfrac{k}{(1-z)^{k+1}} & \text{for } z \leq 0 \end{cases}, \quad \ell''(z) := \begin{cases} \dfrac{k(k+1)}{(1+z)^{k+2}} & \text{for } z \geq 0 \\ \dfrac{k(k+1)}{(1-z)^{k+2}} & \text{for } z \leq 0 \end{cases}$$

and

$$\ell^{-1}(z) = z^{-1/k} - 1, \quad \forall z > 0.$$

From Theorem 5.1 in (Ji & Telgarsky, 2021), we know $\phi(\mathbf{w}) := -\ell^{-1}(\mathcal{L}(\mathbf{w}))$ is convex. Therefore, Condition (A) and (B) in Assumption 5.1 hold. Now we verify Condition (C) in the Assumption 5.1. For $z \geq 0$, it is obvious that $|\ell'(z)| = \frac{k}{(1+z)^k} = k \cdot \ell(z)^{\frac{k+1}{k}}$. For $z \leq 0$, we define

$$h(z) := \frac{|\ell'(z)|}{k \cdot \ell(z)^{\frac{k+1}{k}}} = \frac{2k - \frac{k}{(1-z)^{k+1}}}{k \cdot \left[-2kz + \frac{1}{(1-z)^k}\right]^{\frac{k+1}{k}}}.$$

We know $\lim_{z \to 0^-} h(z) = 1$, and $\lim_{z \to -\infty} h(z) = 0$, and $h(z)$ is continuously differentiable in $(-\infty, 0]$. Moreover, simple computation shows that $h(z)$ is increasing for $z \leq 0$. So $h(z) \leq 1$ for $z \leq 0$, which implies $|\ell'(z)| \leq k \cdot \ell(z)^{\frac{k+1}{k}}$ for $z \leq 0$. Therefore, for any sequence $z_i, 1 \leq i \leq n$, it holds that

$$\begin{aligned}
\frac{1}{n} \sum_{i=1}^{n} |\ell'(z_i)| \cdot \left| (\ell^{-1})' \left( \frac{1}{n} \sum_{i=1}^{n} \ell(z_i) \right) \right| &= \frac{\frac{1}{n} \sum_{i=1}^{n} |\ell'(z_i)|}{k \cdot \left(\frac{1}{n} \sum_{i=1}^{n} \ell(z_i)\right)^{\frac{k+1}{k}}} \leq \frac{k \cdot \frac{1}{n} \sum_{i=1}^{n} \ell(z_i)^{\frac{k+1}{k}}}{k \cdot \left(\frac{1}{n} \sum_{i=1}^{n} \ell(z_i)\right)^{\frac{k+1}{k}}} \\
&\leq \frac{\left(\frac{1}{n} \sum_{i=1}^{n} \ell(z_i)\right) \cdot \max_{1 \leq i \leq n} \ell(z_i)^{1/k}}{\left(\frac{1}{n} \sum_{i=1}^{n} \ell(z_i)\right)^{\frac{k+1}{k}}} \\
&\leq \left( \frac{\max_{1 \leq i \leq n} \ell(z_i)}{\frac{1}{n} \sum_{i=1}^{n} \ell(z_i)} \right)^{\frac{1}{k}} \leq n^{1/k}.
\end{aligned}$$

Therefore, the Condition (C) in Assumption 5.1 is satisfied with $C_\ell = n^{1/k}$. Then, Theorem 5.2 indicates that when $t \geq t_0 := \sqrt{2} n^{1/k} / \gamma^2$, it holds that

$$\mathcal{L}(\overline{\mathbf{w}}_t) = \ell(-\phi(\overline{\mathbf{w}}_t)) \simeq \left(1 + \Theta\left(\gamma^2 \eta t\right)\right)^{-k}. \tag{25}$$

**Example: Probit Negative Log-Likelihood Loss.** We consider the Probit Negative Log-Likelihood Loss, defined as

$$\ell(z) := -\ln\left(F(z)\right), \tag{26}$$

where

$$F(z) := \frac{1}{\sqrt{2\pi}} \int_{-\infty}^{z} \exp\left(-\frac{s^2}{2}\right) \mathrm{d}s$$

is the cumulative density function of standard Gaussian distribution. We also define $f(z) = F'(z)$ as the probability density function of standard Gaussian distribution. Then, we have

$$f'(z) = -z \cdot f(z), \quad \ell'(z) = -\frac{f(z)}{F(z)}, \quad \ell''(z) := \frac{f(z)\left(zF(z) + f(z)\right)}{F(z)^2}.$$

Therefore, we can easily verify the condition A in assumption 5.1 and compute

$$g(z) := \ln\left(\frac{\ell'(z)^2}{\ell(z)\ell''(z)}\right) = \ln\left(f(z)\right) - \ln\left(-\ln\left(F(z)\right)\right) - \ln\left(zF(z) + f(z)\right),$$

which implies

$$g'(z) = -z - \frac{f(z)}{F(z)\ln\left(F(z)\right)} - \frac{F(z)}{zF(z) + f(z)}$$

Lemma G.5 shows that this function is always negative. Therefore, from Lemma G.1, we know $\psi(\cdot)$ is convex and hence, $\phi(\mathbf{w})$ is convex. Next, we verify condition C in Assumption 5.1. From the definition, we know

$$\ell^{-1}(t) = F^{-1}\left(\exp(-t)\right), \quad \left(\ell^{-1}\right)'(t) = -\frac{\exp(-t)}{f\left(F^{-1}\left(\exp(-t)\right)\right)}.$$

We denote $L = \frac{1}{n}\sum_{i=1}^{n} \ell(z_i)$ and $\ell_i = \ell(z_i) = -\ln(F(z_i))$, then

$$\frac{1}{n}\sum_{i=1}^{n}|\ell'(z_i)|\cdot\left|\left(\ell^{-1}\right)'(L)\right| = \frac{1}{n}\sum_{i=1}^{n}\frac{f\left(F^{-1}\left(\exp(-\ell_i)\right)\right)}{\exp(-\ell_i)}\cdot\frac{\exp(-L)}{f\left(F^{-1}\left(\exp(-L)\right)\right)}$$

We define

$$h(z) := \frac{f\left(F^{-1}\left(\exp(-z)\right)\right)}{\exp(-z)}.$$

Then, we can compute

$$h''(z) = \frac{f(x)}{F(x)} + x - \frac{F(x)}{f(x)}, \quad \text{where} \quad x = F^{-1}(\exp(-z)).$$

This quantity is strictly negative for all $z > 0$. So $h$ is concave. Therefore, we know

$$\frac{1}{n}\sum_{i=1}^{n}|\ell'(z_i)|\cdot\left|\left(\ell^{-1}\right)'(L)\right| \leq 1.$$

Therefore, as in the analysis above, we show that when $t \geq t_0 := \sqrt{2}/\gamma^2$, it holds

$$\mathcal{L}(\overline{\mathbf{w}}_t) = \ell\left(-\phi(\overline{\mathbf{w}}_t)\right) \leq \ell\left(\frac{1}{8}\gamma^2\eta t\right) = -\ln\left(F\left(\frac{1}{8}\gamma^2\eta t\right)\right) \leq \mathcal{O}\left(\frac{1}{\gamma^2\eta t}\cdot\exp\left(-c_2\gamma^4\eta^2 t^2\right)\right) \tag{27}$$

where the last inequality comes from Lemma G.4.

# G. Technical Lemmas

## G.1. Convexity of $\psi(\cdot)$

We have the following lemma, which is similar to Lemma 5.2 in (Ji & Telgarsky, 2021). We present the proof here for completeness.

**Lemma G.1.** *If $\frac{\ell'(t)^2}{\ell(t)\cdot\ell''(t)}$ is decreasing on $\mathbb{R}$, then $\psi(\mathbf{z}) := -\ell^{-1}\left(\frac{1}{n}\sum_{i=1}^{n}\ell(z_i)\right)$ is convex.*

*Proof.* For $\mathbf{z} = (z_1, z_2, ..., z_n)^\top \in \mathbb{R}^n$, we have

$$\nabla\psi(\mathbf{z})_i = \frac{-\ell'(z_i)}{n\ell'\left(-\psi(\mathbf{z})\right)}, \quad \nabla^2\psi(\mathbf{z}) := -\text{diag}\left(\frac{\ell''(z_1)}{n\ell'(-\psi(\mathbf{z}))}, \frac{\ell''(z_2)}{n\ell'(-\psi(\mathbf{z}))}, ..., \frac{\ell''(z_n)}{n\ell'(-\psi(\mathbf{z}))}\right) + \frac{\ell''(-\psi(\mathbf{z}))}{\ell'(-\psi(\mathbf{z}))}\nabla\psi(\mathbf{z})\psi(\mathbf{z})^\top.$$

It suffices to show that for any $\mathbf{v} \in \mathbb{R}^n$,

$$-\sum_{i=1}^{n}\frac{\ell''(z_i)}{n\ell'(-\psi(\mathbf{z}))}v_i^2 \geq -\frac{\ell''(-\psi(\mathbf{z}))}{\ell'(-\psi(\mathbf{z}))}\left(\sum_{i=1}^{n}\frac{-\ell'(z_i)}{n\ell'\left(-\psi(\mathbf{z})\right)}v_i\right)^2.$$

Recall that $\ell'(z) < 0$ for any $z \in \mathbb{R}$. Combined with Cauchy-Schwarz inequality, this gives

$$\left( \sum_{i=1}^{n} \frac{-\ell'(z_i)}{n\ell'(-\psi(\mathbf{z}))} v_i \right)^2 \leq \left( \sum_{i=1}^{n} \frac{-\ell''(z_i)}{n\ell'(-\psi(\mathbf{z}))} v_i^2 \right) \left( \sum_{i=1}^{n} -\frac{(-\ell'(z_i))^2}{n\ell'(-\psi(\mathbf{z}))\,\ell''(z_i)} \right).$$

So it suffices to prove

$$\sum_{i=1}^{n} \frac{(-\ell'(z_i))^2}{n\ell''(z_i)} \leq \frac{(-\ell'(-\psi(\mathbf{z})))^2}{\ell''(-\psi(\mathbf{z}))},$$

which is equivalent to

$$\sum_{i=1}^{n} \frac{(-\ell'(z_i))^2}{n\ell''(z_i)} \leq \frac{\left(-\ell'\left(\ell^{-1}\left(\frac{1}{n}\sum_{i=1}^{n}\ell(z_i)\right)\right)\right)^2}{\ell''\left(\ell^{-1}\left(\frac{1}{n}\sum_{i=1}^{n}\ell(z_i)\right)\right)}$$

We define $h(z) = \frac{1}{n}\ell(z)$. This is equivalent to

$$\sum_{i=1}^{n} \frac{\left(-h'\left(h^{-1}\left(h\left(z_i\right)\right)\right)\right)^2}{h''(h^{-1}\left(h\left(z_i\right)\right))} \leq \frac{\left(-h'\left(h^{-1}\left(\sum_{i=1}^{n}h(z_i)\right)\right)\right)^2}{h''\left(h^{-1}\left(\sum_{i=1}^{n}h(z_i)\right)\right)}$$

We consider the function $g : (0, \infty) \to \mathbb{R}$ :

$$g(s) := \frac{\left(-h'\left(h^{-1}(s)\right)\right)^2}{h''\left(h^{-1}(s)\right)}.$$

Then,

$$\frac{g(s)}{s} = \frac{(h'(t))^2}{h(t)h''(t)} \quad \text{where} \quad t = h^{-1}(s).$$

Since $h^{-1}(s)$ is decreasing and $\frac{(h'(t))^2}{h(t)h''(t)} = \frac{(\ell'(t))^2}{\ell(t)\ell''(t)}$ is decreasing in $t$, we know $g(s)/s$ is increasing on $s \in (0, \infty)$. This indicates for any $a, b > 0$, we have $\frac{g(a+b)}{a+b} \geq \frac{g(a)}{a}$ and $\frac{g(a+b)}{a+b} \geq \frac{g(b)}{b}$, which implies

$$a \cdot g(a+b) \geq (a+b) \cdot g(a), \quad b \cdot g(a+b) \geq (a+b) \cdot g(b).$$

Adding these two equations and normalizing it, we have $g(a+b) \geq g(a) + g(b)$, which shows $g(\cdot)$ is super-additive on $(s, \infty)$. This gives

$$\sum_{i=1}^{n} g(h(z_i)) \leq g\left( \sum_{i=1}^{n} h(z_i) \right),$$

which concludes the proof. $\qquad \square$

## G.2. Lemmas Regarding Gaussian Distribution and Mill's Ratio

**Lemma G.2** (Proposition 1.1 in (Mukherjee, 2016))**.** *Let $X$ be a random variable with distribution function $F$. Set $m_k(x) = \mathbb{E}\left[X^k\mathbb{I}(X > x)\right]$. Assume $\mathbb{E}\left|X\right|^N < \infty$. Then, for all $0 \leq n \leq N$, we have*

$$\sum_{k=0}^{n} C_n^k m_k(x) (-x)^{n-k} \geq 0.$$

*Proof.* Note that the left hand side is just $\mathbb{E}(X - x)_+^n$, where $(a)_+ := \max\{a, 0\}$. $\qquad \square$

**Lemma G.3** (Proposition 1.2 in (Mukherjee, 2016))**.** *Suppose $\mathbb{E}\left|X\right|^{k+1} < \infty$. Define $m_k(x) = \mathbb{E}\left[X^k\mathbb{I}(X > x)\right]$. Then,*

$$m_{k+1}(x) = xm_k(x) + \int_x^\infty m_k(u)\mathrm{d}u.$$

*Proof.* Note that

$$\int_x^\infty m_k(u)\mathrm{d}u = \int_x^\infty \mathbb{E}\left(X^k \mathbb{I}(X > u)\right)\mathrm{d}u = \mathbb{E}\left[X^k \int_x^\infty \mathbb{I}(X > u)\mathrm{d}u\right] = m_{k+1}(x) - xm_k(x).$$

$\square$

**Lemma G.4.** *We define $F(z)$ and $f(z)$ as the cumulative density function and probability density function for one-dimensional standard Gaussian distribution. Then,*

- *For any $z > 0$, one has*

$$\frac{zf(z)}{z^2+1} \leq 1 - F(z) \leq \frac{f(z)}{z}. \tag{28}$$

- *For any $z \in \mathbb{R}$, one has*

$$z \cdot F(z) + f(z) \geq 0. \tag{29}$$

- *For any $z > 0$, one has*

$$f(z)\left(\frac{1}{z} - \frac{1}{z^3}\right) \leq 1 - F(z) \leq f(z)\left(\frac{1}{z} - \frac{1}{z^3} + \frac{3}{z^5}\right). \tag{30}$$

- *For any $z > 0$, one has*

$$\frac{z^3 + 5z}{z^4 + 6z^2 + 3} \leq \frac{1 - F(z)}{f(z)} \leq \frac{z^2 + 2}{z(z^2 + 3)}. \tag{31}$$

*Proof.* The first inequality comes from the standard Mill's ratio for Gaussian distribution. To see the second claim, the first inequality gives $zF(z) + f(z) \geq z > 0$ for any $z > 0$. For $z = 0$, this is obvious. For $z < 0$, we apply the first claim on $-z$ to get

$$F(z) = 1 - F(-z) \leq \frac{f(-z)}{-z} = \frac{f(z)}{-z},$$

which implies the second claim directly. The third inequality comes from Exercise 2.2 in (Wainwright, 2019). The last inequality is a refinement of the classical Mill's ratio and it comes from a notes (Mukherjee, 2016). However, the results from that notes are incorrect. So we give the correct version here for completeness. Lemma G.3 gives

$$m_0(z) = 1 - F(z), \quad m_1(z) = f(z), \quad m_2(z) = zf(z) + (1 - F(z)),$$
$$m_3(z) = z^2 f(z) + 2f(z), \quad m_4(z) = x^3 f(z) + 3zf(z) + 3(1 - F(z)).$$

Taking $n = 3$ in Lemma G.2 gives the right side of (31), while $n = 4$ gives the left side. $\square$

The following result shows a standard result for standard Gaussian distribution.

**Lemma G.5.** *We define $F(z)$ and $f(z)$ as the cumulative density function and probability density function for one-dimensional standard Gaussian distribution. Then, for any $z \in \mathbb{R}$, one has*

$$z + \frac{f(z)}{F(z)\ln(F(z))} + \frac{F(z)}{zF(z) + f(z)} \geq 0. \tag{32}$$

*Proof.* The claim holds trivially for $z = 0$. Let's then consider $z > 0$. We write $F$ and $f$ for $F(z)$ and $f(z)$, respectively. The claim is equivalent with

$$\left(z^2 + 1\right)F^2\ln F + zFf\ln F + zFf + f^2 \leq 0. \tag{33}$$

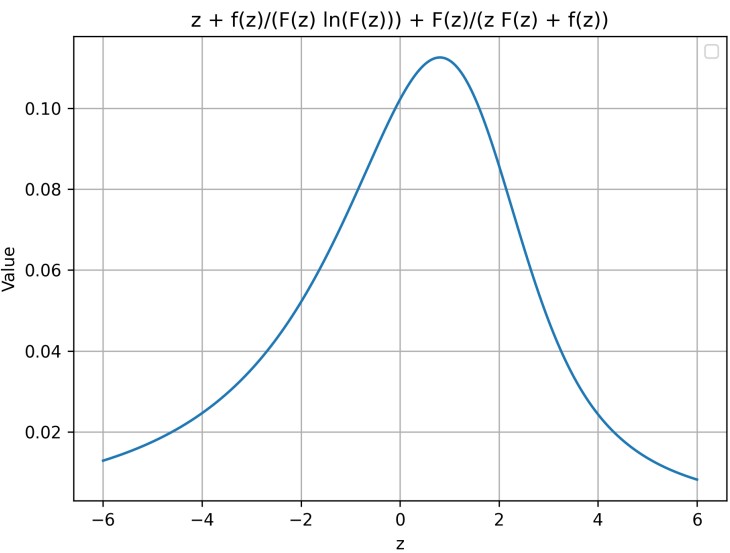

*Figure 2.* The Objective function in Lemma G.5 when $-5 \leq z \leq 2$.

Since $\ln F \leq F - 1$ for any $F = F(z)$, when $z \geq 2$, we have

$$
\left(z^2 + 1\right) F^2 \ln F + zFf \ln F + zFf + f^2 \leq \left(z^2 + 1\right) F^2(F - 1) + zFf(F - 1) + zFf + f^2
$$

$$
= f \cdot \left(\left(z^2 + 1\right) F^2 \frac{F - 1}{f} + zF^2 + f\right) \leq f \cdot \left(f - \frac{2F^2}{z^3 + 6z + 3/z}\right) \qquad \text{(From (31))}
$$

$$
\leq f \cdot \left(f - \frac{2 \cdot 0.97^2}{z^3 + 6z + 3/z}\right) \qquad \text{($F \geq 0.97$ when $z \geq 2$)}
$$

$$
\leq f \cdot \left(\frac{1}{\sqrt{2\pi}} \cdot \frac{1}{1 + \frac{1}{2}z^2 + \frac{1}{8}z^4} - \frac{2 \cdot 0.97^2}{z^3 + 6z + 3/z}\right) \qquad \text{($\exp(-\frac{1}{2}z^2) \leq \frac{1}{1 + \frac{1}{2}z^2 + \frac{1}{8}z^4}$)}
$$

$$
< 0 \qquad \text{(From numerical computation)}
$$

Therefore, the claim holds for $z \geq 2$. Next, we will deal with the case when $z < -5$. We denote $x = -z$. Since $F(-z) = 1 - F(z)$ and $f(-z) = f(z)$, (33) is equivalent with

$$
\left(x^2 + 1\right)(1 - F(x))^2 \ln(1 - F(x)) - x(1 - F(x)) f(x) \ln(1 - F(x)) - x(1 - F(x)) f(x) + f(x)^2 \leq 0. \qquad (34)
$$

Now we write $F$ and $f$ for short of $F(x)$ and $f(x)$, respectively. We denote $m(x) := \frac{1 - F(x)}{f(x)}$. This is equivalent with

$$
(x^2 + 1)m(x)^2 \ln(1 - F(x)) - x \cdot m(x) \ln(1 - F(x)) - x \cdot m(x) + 1 \leq 0.
$$

When $z \leq -5$, and $x \geq 5$, from (31), one has

$$
\ln(1 - F(x)) \leq \ln\left(\frac{f(x)}{x}\right) = -\frac{1}{2}\ln(2\pi) - \frac{1}{2}x^2 - \ln(x),
$$

which implies

$$(x^2 + 1)m(x)^2 \ln(1 - F(x)) - x \cdot m(x) \ln(1 - F(x)) - x \cdot m(x) + 1$$

$$\leq m(x) \cdot \frac{(x^2 + 1)(x^3 + 5x)}{x^4 + 6x^2 + 3} \ln(1 - F(x)) - x \cdot m(x) \ln(1 - F(x)) - x \cdot m(x) + 1$$

$$= \frac{2x}{x^4 + 6x^2 + 3} \cdot m(x) \cdot \ln(1 - F(x)) - x \cdot m(x) + 1$$

$$\leq \left( \frac{-x^2 - 2\ln(x) - \ln(2\pi)}{x^4 + 6x^2 + 3} - 1 \right) x \cdot m(x) + 1 \tag{35}$$

$$\leq \left( \frac{-x^2 - 2\ln(x) - \ln(2\pi)}{x^4 + 6x^2 + 3} - 1 \right) \cdot \frac{x^4 + 5x^2}{x^4 + 6x^2 + 3} + 1 \qquad \text{(From (31) in Lemma G.4)}$$

$$= \frac{(4 - \ln(2\pi) - 2\ln(x)) x^4 + (21 - 5\ln(2\pi) - 10\ln(x)) x^2 + 9}{(x^4 + 6x^2 + 3)^2}$$

$$\leq 0 \qquad \text{(When } x \geq 5 \text{ from numerical computation)}$$

Therefore, we have proven the claim for $z \leq -5$ and $z \geq 2$. For $-5 \leq z \leq 2$, we numerically show the claim holds in Figure 2. $\qquad \square$

