# OpenReview forum: "Gradient Descent Converges Arbitrarily Fast for Logistic Regression via Large and Adaptive Stepsizes"
_ICML.cc/2025/Conference — ICML 2025 poster_

### Official Review · Reviewer_QkHV · 2025-03-12

**Overall Recommendation:** 1

**Summary:**

The paper investigates the convergence of gradient-based methods with large and adaptive step sizes on logistic regression with linearly separable data. The main result establishes that GD can achieve arbitrarily fast convergence rates by using an adaptive step-size schedule.
Furthermore, the authors prove a lower bound on the iteration complexity for any first-order method. The results are extended to a broader class of loss functions and two-layer neural networks.

**Claims And Evidence:**

See Summary.

**Essential References Not Discussed:**

Theoretical analysis on the implicit bias and convergence of logistic regression:

[1] Ji et al. Fast Margin Maximization via Dual Acceleration. (ICML 2022)

[2] Wang et al. On accelerated perceptrons and beyond. (ICLR 2023)

[3] Wang et al. Achieving margin maximization exponentially fast via progressive norm rescaling. (ICML 2024)

**Experimental Designs Or Analyses:**

No experiments.

**Methods And Evaluation Criteria:**

N/A

**Other Comments Or Suggestions:**

See Weaknesses.

**Other Strengths And Weaknesses:**

**Strengths.**
- This paper proves that GD can achieve arbitrarily fast convergence rates via large stepsizes, surpassing previously established rates.

**Weaknesses.**

I have two primary concerns regarding the analysis in the setting of linear regression on linearly separable data.

- **The arbitrarily fast convergence rate is trivial in this setting.** For simplicity, consider the exp-loss $\ell(z)=e^{-z}$ and let $L(w)=\frac{1}{n}\sum_{i=1}^n\ell(y_i f(w;x_i))$. Let {$w_t$} be trained by GD with either a constant or adaptive step size. The proof proceeds as follows:
  - *Stage I:  Correct classification*. Prior works [Soudry et al. (2018); Ji & Telgarsky (2021)] establish that there exists a time $T_0$ such that $L(w_{T_0})<\frac{1}{n}$, ensuring all data points are correctly classified: $\min_{i\in[n]}y_i f(w_{T_0};x_i) >0$.
  - *Stage II: Naively scaling the parameter norm.* Notice thatthe linear model satisfies $f(Cw;x)=Cf(w;x)$ for any $C>0$, implying that $\min_{i\in[n]} y_i f(C w_{T_0};x_i) = C \min_{i\in[n]} y_i f(w_{T_0};x_i)$. Therefore, increasing the norm $C\to+\infty$ can lead to $\min_{i\in[n]} y_i f(C w_{T_0};x_i) \to+\infty$, implying $L(Cw_{T_0})\to 0$.


- **Unclear connection to Edge of stability (EoS).** Although this work frequently references EoS, it does not establish a clear link to it. For GD, EoS typically refers to the phenomenon where $\lambda_{\max}(\nabla^2 L(w_t))$ oscillates around $\frac{2}{\eta_t}$. However, this article does not analyze $\lambda_{\max}(\nabla^2 L(w_t))$ or its relationship with $2/\eta_t$.

**Questions For Authors:**

Can the proposed step size find the max-margin classifier in the logistic regression setting, similar to standard GD? Additionally, what is the max-margin rate?

Notably, while arbitrarily fast convergence can be trivially achieved in linear regression on linearly separable data (see Weaknesses), fast margin maximization is not as straightforward.

**Relation To Broader Scientific Literature:**

See Summary.

**Theoretical Claims:**

Partially.

---

> ### Author Rebuttal · Authors · 2025-03-31
>
> Thank you for your comments and pointing out missing references. We will cite and discuss them in the revision. We address your questions below.
>
> ---
>
> Q1: “The arbitrarily fast convergence rate is trivial in this setting…..”
>
> A1: We respectfully disagree. Note that the algorithm you proposed is much slower than ours. This is because in Stage I, GD needs to attain a risk smaller than $\Theta(1/n)$, which requires $O(n/\gamma^2)$ steps for a constant stepsize or $O(\ln(n)/\gamma^2)$ for small adaptive stepsizes. In comparison, we show that GD with large adaptive stepsizes only needs $\Theta(1/\gamma^2)$ steps.
>
> Moreover, we can improve our lower bounds construction to show that $\Theta(1/\gamma^2)$ steps is minimax optimal for any first-order batch method to find a linear separator for a separable dataset with margin $\gamma$. The proof is provided at the end of our response. Thus our algorithm is minimax optimal, while the algorithm you proposed is suboptimal by a $n$ or $\ln (n)$ factor. We hope this clarifies your concern!
>
> ---
>
> Q2: “Unclear connection to Edge of stability (EoS). Although this work frequently references EoS, it does not establish a clear link to it. For GD, EoS typically refers to the phenomenon where $\lambda_{max}(\nabla^2 L(w_t))$ oscillates around $2/\eta_t$. However, this article does not analyze $\lambda_{max}(\nabla^2 L(w_t))$ or its relationship with $2/\eta_t$.”
>
> A2: This seems to be a misunderstanding. Note that [Cohen et al., 2020] described EoS by
>
> “...gradient descent enters a regime we call the Edge of Stability, in which (1) the sharpness hovers right at, or just above, the value $2/\eta$; and (2) the train loss behaves nonmonotonically over short timescales, yet decreases consistently over long timescales….”
>
> The second bullet is exactly our definition of EoS, and is arguably more fundamental than the first bullet, since a key surprising feature of EoS is its inconsistency with the descent lemma (see their abstract). We believe our references to EoS are justified.
>
> ---
>
> We formally define first-order batch methods: Let $\ell(\cdot)$ be a locally Lipschitz function.
> We say $w_t$ is the output of a first-order batch method in $t$ steps with initialization $w_0$ on dataset $(x_i, y_i)\_\{i=1\}\^\{n\}$, if it can be generated by $w_{k} \in w_0 +Lin \\\{\nabla L(w_0),  \dots, \nabla L(w_{k-1})\\\},k= 1,\dots, t,$
> where $Lin$ is the linear span of a vector set and $+$ is the Minkowski addition.
>
> The improved lower bound: For every $0 < \gamma < 1/6$, $n>16$, and $w_0$,
> there exists a dataset $(x_i,y_i)\_\{i=1\}\^n$ satisfying Assumption 1.1 such that the following holds.
> For any $w_t$ output by a first-order batch method in $t$-steps with initialization $w_0$ on this dataset, we have
> $\min_{i\in[n]} y_i x_i^\top  w_t> 0 $ implies that $t \geq \min\\\{\ln(n)/(8 \ln 2), 1/(30 \gamma^2)\\\}.$
>
> Proof: We define $d := \lfloor 1/5\gamma^2 \rfloor \geq 6,$ where $\lfloor \cdot \rfloor$ is the floor function. Let $(e_i)\_\{i=1\}\^d$ be a set of standard basis vectors. Note that all defined first order methods defined are rotational invariants. Therefore, we can without loss of generality assume $w_0$ is propotional to $e_1$. Let $k := \min\\{\lfloor \log_2 n \rfloor, d-2\\} \geq 4.$ We construct $(x_i, y_i)\_\{i=1\}\^n$ as follows. Let $y_i=1$ for all $i\in[n]$. For $j=1,\dots,k$, let $x_i := (2/\sqrt{5})e_{j+1} - (1/\sqrt{5}) e_{j+2}$ for $2^{k}-2^{k-j+1}+1 \leq i \leq 2^{k}-2^{k-j}$. Let the remaining $x_i$'s be $x_i:= (1/\sqrt{5}) e_{k+2}$ for $2^k \le i\le n$. Note that $\|x_i\|\le 1$ for $i=1,\dots,n$. Moreover, for the unit vector $w^* = (1/\sqrt{d})  (1,1,\dots,1)^{\top}$, we have $y_i x_i^\top w^* \geq \gamma$ for every $i$. Thus, the dataset satisfies Assumption 1.1.
>
> For a vector $w$, we also write it as $w := (w^{(1)}, w^{(2)},\dots,w^{(d)})^\top$.
> Then, the objective function can be written as
> $$
>     L(w)
>     = \frac{1}{n} \Bigg[\sum\_\{j=1\}\^k 2^{k-j}   \ell\bigg(\frac{2}{\sqrt{5}} w^{(j+1)} - \frac{1}{\sqrt{5}} w^{(j+2)}\bigg) + \big(n-2^{k}+1\big)  \ell\bigg(\frac{1}{\sqrt{5}} w^{(k+2)}\bigg)\Bigg].
> $$
> Consider a sequence $(w_s)\_\{s=0\}\^t$ generated by a first-order method. We know the gradient at $w_0$ vanishes in all coordinates except the second and the $(k+2)$-th coordinates. By induction, we conclude that for $t \leq t_0-2$ for $t_0:= \lfloor (k+1)/2\rfloor,$ it holds that $w_t \in Lin\\{e_1, \dots, e_{t+1}, e_{k+3-t}, \dots, e_{k+2}\\}$. So for all $(w_s)_\{s=0\}\^t$, their $t_0$-th and $(t_0+1)$-th coordinates must be zero. By our dataset construction, there exists $i\le 2^k-1$ such that $y_i x_i^\top w_k= 0$ for $k=0,\dots,t.$ This means that the dataset cannot be separated by any of $(w_k)\_\{k=0\}\^t$. Thus, for the first-order method to output a linear separator, we must have $t \geq t_0-1 \geq \lfloor(k-1)/2\rfloor\geq \min\\{\ln n/(8 \ln 2), 1/(30 \gamma^2)\\}$.
>
> We will elaborate on this lower bound and its proof in the revision.

---

### Official Review · Reviewer_YHGM · 2025-03-14

**Overall Recommendation:** 4

**Summary:**

This paper considers using GD to optimize linear classification losses, primarily the exp and logistic losses, but also extended to certain qualitatively similar losses. They show that GD using a particular adaptive stepsize schedule which is roughly proportional to the reciprocal of the loss value (for the logistic and exp losses at least) can converge arbitrarily fast on realizable problems after a short, margin-dependent burn in period. The loss typically does not decrease monotonically while using their schedule, and in fact, they show that if the stepsizes are such that the loss does decrease monotonically, then the rate of convergence is necessarily slower. They extend these results to training the bottom layer of a two layer network with leaky relu activations which has similar behavior. Finally, they show qualitatively similar results for a broader class of losses satisfying certain conditions.

**Claims And Evidence:**

Yes, the claims are well supported by the evidence they provide in the form of theoretical analysis.

**Essential References Not Discussed:**

None that I know of.

**Experimental Designs Or Analyses:**

N/A

**Methods And Evaluation Criteria:**

This is a theory paper, and they address the questions they consider through theoretical analysis, so yes.

**Other Comments Or Suggestions:**

pg 3: "less as less effective" -> "less and less effective"
eq (13): what is (-\ell^{-1}(z))' referring to? What is z? Is this supposed to be the derivative of the inverse of negative ell evaluated at 1/n sum_i \ell(z_i)? If so, the "(z)" is a little confusing here.

**Other Strengths And Weaknesses:**

Strengths: This is a very nicely written paper, which clearly explains the why and the how. The proof sketches (and proofs themselves) are nicely written and give a solid intuition of how the results were established. Of the papers I am reviewing in ICML this year, this is easily the most readable.

Weaknesses: This is an interesting set of results, but it does feel a little bit like it runs the risk of "overfitting" to separable logistic/exp loss classification. These problems are a little bit unusual in that reducing the loss from a very small value to substantially smaller value requires using an extremely large stepsize. This doesn't invalidate this (very nice) paper, but I would like to see some evidence that this type of approach / mode of analysis can tell us something about other more difficult problems. I don't mean to say that separable linear classification is an unimportant problem---it's not---but I would say that at this point, we are not desperately in need of new methods for solving separable linear classification problems. Is this type of adaptive stepsize useful for non-separable linear classification problems? Does it help when training both layers of a 2 layer MLP (in theory or practice)? Does it help when training more realistic neural networks in practice?

**Questions For Authors:**

See above.

**Relation To Broader Scientific Literature:**

The main contribution of this paper seems to be to extend the results of Ji and Telgarsky 2021, who studied the same adaptive stepsizes, but using a much smaller \eta such that the convergence was monotonic. Due to the monotonicity, their result was weaker, and this paper shows that you still have fast convergence, even when \eta is chosen much larger than what would be needed for stability, and in fact the larger \eta is the faster convergence is without limit.

**Theoretical Claims:**

I read through the proofs and they appear to be accurate as best I can tell. The only exception is what I believe is simply a typo in the statement of Theorem 5.2: I think the rhs of the last displayed equation should be \mathcal{L}(\bar{w}_t) \leq \ell(\frac{1}{8}\gamma^2 \eta t), i.e. without the minus sign (otherwise, it gets worse with larger t since \ell is assumed to be decreasing).

---

> ### Author Rebuttal · Authors · 2025-03-31
>
> Thank you for supporting our paper! You are correct that there is a typo in the statement of Theorem 5.2. We will make sure to fix it (and all other typos) in the revision. We address your other questions as follows.
>
> ---
> Q1: “...Is this type of adaptive stepsize useful for non-separable linear classification problems? Does it help when training both layers of a 2 layer MLP (in theory or practice)? Does it help when training more realistic neural networks in practice?”
>
> A1: Good question. Our main focus is understanding the benefits of EoS/large stepsizes. It is unclear to what extent our results generalize to other cases, such as non-separable linear classification, two-layer networks with non-linearly separable data, or even practical network training. We will comment on this as a future direction.
>
> ---
> Q2: “eq (13): what is $(-\ell^{-1}(z))'$ referring to? What is z? Is this supposed to be the derivative of the inverse of negative ell evaluated at 1/n sum_i \ell(z_i)? If so, the "(z)" is a little confusing here.”
>
> A2: Here $(-\ell^{-1}(z))$ should be replaced by $-\ell^{-1}$. This is a typo and we will fix it in the revision.

---

### Official Review · Reviewer_VFUd · 2025-03-14

**Overall Recommendation:** 4

**Summary:**

The paper shows that in logistic regression with linearly separable data, gradient descent can achieve arbitrarily fast convergence through large and adaptive stepsizes for exponential and logistic loss. This occurs in the edge of stability regime and does not require monotonic risk decrease to occur. Additionally, lower bounds for stable regime adaptive stepsize GD convergence is established, along with a general lower bound for the number of burn-in steps. Finally, with additional assumptions on the loss allows for an improved convergence rate for other losses.

**Claims And Evidence:**

Yes, overall the claims made here seem to be supported by sufficient proofs.

**Essential References Not Discussed:**

N/A

**Experimental Designs Or Analyses:**

N/A

**Methods And Evaluation Criteria:**

N/A

**Other Comments Or Suggestions:**

Lines 269-270 - Spacing for ".Specifically" is messed up

Line 272-273 - Missing period before "However, ..."

**Other Strengths And Weaknesses:**

Overall, the paper is well developed, concise, and sufficiently rigorous for a theory paper. It builds on an existing body of work and provides novel and significant results.

**Questions For Authors:**

N/A

**Relation To Broader Scientific Literature:**

These contributions are related to analysis of gradient descent and logistic regression. To the best of my understanding, these results improve on the existing literature beyond step size scheduling and monotonicity assumptions.

**Theoretical Claims:**

I checked the proofs of the theoretical claims to the best of my ability and found no significant issues.

---

> ### Author Rebuttal · Authors · 2025-03-31
>
> Thank you for supporting our paper! We will make sure to correct all the typos in the revision.

---

### Official Review · Reviewer_xEgy · 2025-03-14

**Overall Recommendation:** 3

**Summary:**

The authors analyze the gradient descent optimization procedure for logistic regression in the large-stepsize regime. Their upper bounds lead to a "soft-perceptron" view of logistic regression, which extends to two-layer leaky ReLU networks and other loss functions with regularity properties similar to exponential and logistic losses. Since their upper bounds are for average-iterate or best-iterate convergence, both in large-stepsize regimes, they also provide a lower bound in the stable regime, manifesting that large stepsizes is essential for arbitrarily fast convergence after the burn-in stage. They also provide a lower bound manifesting that the burn-in stage is necessarily if the last-iterate or the average-iterate convergence is well.

**Claims And Evidence:**

See the Section "Theoretical Claims".

**Essential References Not Discussed:**

To be best of my knowledge, most essential references are discussed.

**Experimental Designs Or Analyses:**

The intuition behind the simulation is coherent with the upper bounds.

**Methods And Evaluation Criteria:**

Not quite applicable for this theoretical paper. The only subtlety is that while Assumption 1.1 did not model the randomness of the data, the authors random sample features from the unit hypersphere to do simulations. This subtlety leads to a weakness about Assumption 1.1. See Section "Other Strengths And Weaknesses" for details.

**Other Comments Or Suggestions:**

See the Section "Theoretical Claims" for the comments on typos in the paper.

**Other Strengths And Weaknesses:**

### On the Lower Bounds

The evidence provided by simulations and upper bounds actually motivates the following two intuitive points:
1. For the "large-stepsize" regime, the last-iterate convergence is not good in general, even with high burn-in cost.
2. For the "large-stepsize" regime, if the burn-in cost is not high enough, the average-iterate convergence is also not good.

However, Theorem 3.2 only manifest the second point above. This might not be fatal, but the discussion about the first point I mentioned here is indeed missing in the main text.

### Other Subtleties
- For linearly separable binary classification, the burn-in cost in this paper corresponds to the exact iteration complexity of perceptron on the very same task. This connection between logistic regression and "soft perceptron" is interesting. However, perceptron is able to find the separating hyperplane in the "last-iterate" sense after $1/\gamma^2$ iterations, while the simulations and upper bounds in this paper intuitively suggest an evidence that logistic regression is only able to do so in an "average-iterate" or even "best-iterate" sense. This subtlety make logistic regression sounds like a method even worse than perceptron, at least for this task. The authors should elaborate more on this point.
- As the authors did in the simulations, sampling data in real-world scenarios often involves randomness. Even in the i.i.d. case, if the support of the data distribution is decently non-trivial, i.e., regions near the boundary are not null sets, $\gamma$ will decrease as the sample size $n$ increases, which will in turn amplify the burn-in cost $\gamma^{-2}$ in this paper. Thus, Assumption 1.1 has a significant oversight even for the optimization results in this paper.
- By the way, $y_i = 1$ in Assumption 1.1 is mathematically not a typo since all proofs go through even all labels are positive, but it makes the readers realize that the implication of the upper bounds (when all labels are positive) is not very interesting.

**Questions For Authors:**

See "Theoretical Claims" and "Other Strengths And Weaknesses".

**Relation To Broader Scientific Literature:**

This paper falls into the line of "large-stepsize gradient descent for linearly separable logistic regression" papers. Direct predecessors of this paper include
- Wu, Jingfeng, et al. "Large Stepsize Gradient Descent for Logistic Loss: Non-Monotonicity of the Loss Improves Optimization Efficiency." The Thirty Seventh Annual Conference on Learning Theory. PMLR, 2024.
- Cai, Yuhang, et al. "Large stepsize gradient descent for non-homogeneous two-layer networks: Margin improvement and fast optimization." Advances in Neural Information Processing Systems 37 (2024): 71306-71351.

**Theoretical Claims:**

The Theorems stated in the paper are correct and the proofs are largely correct. However, several point are worth noting.

- It is widely accepted that the average-iterate convergence and the last-iterate convergence are not directly comparable in general. Thus, the authors' comment that "Theorem 2.2 improves Proposition 2.1" is not accurate, since Theorem 2.2 is not about the last-iterate convergence.
- Clarity: In the proof of Theorem 3.2, around the $\geq$ on Line 733-734, the implicit assumption $\ell(0) = 1$ should be explicitly stated in the proof, as well as in the statement of this theorem, which is in the main text.
- Math typo: The numerator and the denominator in Equation (16) are upside down, though the consequence of which is not fatal.
- Minor math typo: The $\leq$ in Line 708-709 should be $\asymp$ to make the margin condition go smoothly.
- Typo on Line 981-982: $C_{\ell}$ -> $C_{\ell}^2$ (as well the subsequent typos derived from it)
- Minor typo on Line 184: "Theorem 5.2" -> "Theorem 2.2"
- Minor typo in the last sentence of Section 5: "last" -> "penultimate"
- Minor typo on Line 951: "Assumption D.1" -> "Assumption 5.C"
- Minor comment: The 1st equation block of the statement of Theorem 3.1 is *in a verbatim way* highly similar to that in [1, Theorem 3]. In this case, instead of saying "movinated by", I would suggest saying "following the construction". Otherwise, the "edit distance" between the two equations is too small.

References

[1] Wu, Jingfeng, et al. "Large Stepsize Gradient Descent for Logistic Loss: Non-Monotonicity of the Loss Improves Optimization Efficiency." The Thirty Seventh Annual Conference on Learning Theory. PMLR, 2024.

---

> ### Author Rebuttal · Authors · 2025-03-31
>
> We thank the reviewer for their comments and for pointing out the typos. We will make sure to fix all typos in the revision. We address your questions as follows.
>
> ---
> Q1: “It is widely accepted that the average-iterate convergence and the last-iterate convergence are not directly comparable in general. Thus, the authors' comment that "Theorem 2.2 improves Proposition 2.1" is not accurate, since Theorem 2.2 is not about the last-iterate convergence.”
>
> A1: When treating the output as part of the algorithm design, it is fair to compare GD with large adaptive stepsizes that outputs the averaged iterate and GD with small adaptive stepsizes that outputs the last iterate. Moreover, GD in Proposition 2.1 satisfies the descent lemma, so their averaged iterate would only be slower than their last iterate—so Theorem 2.2 improves Proposition 2.1. We will clarify this in the revision.
>
> ---
> Q2: “The evidence provided by simulations and upper bounds actually motivates the following two intuitive points: 1. For the "large-stepsize" regime, the last-iterate convergence is not good in general, even with high burn-in cost. 2. … However, Theorem 3.2 only manifest the second point above. This might not be fatal, but the discussion about the first point I mentioned here is indeed missing in the main text.”
>
> A2: We would like to point out that our Figure 1 does not seem to imply that “the last-iterate convergence is not good in general”. Our paper is limited to averaged iterate, and it remains open to analyze the performance of the last-iterate. We will comment on this in the revision.
>
> ---
> Q3: “For linearly separable binary classification, …perceptron is able to find the separating hyperplane in the "last-iterate" sense after 1/\gamma^2 iterations, while the simulations and upper bounds in this paper intuitively suggest an evidence that logistic regression is only able to do so in an "average-iterate" or even "best-iterate" sense. This subtlety make logistic regression sounds like a method even worse than perceptron, at least for this task. The authors should elaborate more on this point.”
>
> A3: When viewing output design as part of the algorithm, our algorithm (GD with large stepsizes and outputs the averaged iterate) matches perceptron in terms of step complexity. So it seems unfair to say our algorithm is “worse” than perceptron. Additionally, we have improved our lower bound construction, showing that our algorithm is minimax optimal among all first-order methods in this problem (see our response to Reviewer QkHV). We will make a detailed comparison with Perceptron in the revision.
>
> ---
> Q4: “.... Even in the i.i.d. case, if the support of the data distribution is decently non-trivial, i.e., regions near the boundary are not null sets, \gamma will decrease as the sample size n increases, which will in turn amplify the burn-in cost $\gamma^{−2}$ in this paper. Thus, Assumption 1.1 has a significant oversight even for the optimization results in this paper.”
>
> A4: This is a good question. First, we would like to emphasize that this is a standard assumption in binary classification problems and has been widely adopted in literature. We agree that under certain distributional assumptions, the margin of the empirical dataset might be a decreasing function of the sample size. However there are also other cases where the margin of the empirical data remains large. For example, this is the case if the population distribution is (almost) separable with a margin $\gamma$. We will discuss this in the revision.
>
> ---
> Q5: “By the way, yi=1 in Assumption 1.1 is mathematically not a typo since all proofs go through even all labels are positive, but it makes the readers realize that the implication of the upper bounds (when all labels are positive) is not very interesting.”
>
> A5: Note that our results apply to the case where $y_i \in \{\pm 1 \}$. In this case, we can replace $y_i$ by $1$ and $x_i$ by $y_i x_i$ respectively, then apply our current analysis. So our assumption that $y_i=1$ does not cause any loss of generality. We will clarify this in the revision.

---

### Decision · Program_Chairs · 2025-05-01

**Decision:**

Accept (poster)

**Comment:**

This paper presents a theoretical study on the convergence behavior of gradient descent with large adaptive stepsizes for logistic regression in the linearly separable regime. The central result is that arbitrarily fast convergence can be achieved after the burn-in stage, without requiring monotonic loss decrease. The analysis further generalizes to certain two-layer neural network settings and other loss functions.

Reviewers acknowledged the novelty, clarity, and theoretical strength of the paper. While there is a concern regarding the significance of the main result (arguing that fast convergence may be trivial), this paper’s value lies in its explicit and rigorous characterization of how large-step GD operates in this setting.

Overall, this paper makes a clear, technically solid contribution to the theoretical understanding of gradient dynamics under large stepsizes. I recommend acceptance.